# Interlayer magnetophononic coupling in MnBi$_2$Te$_4$

Hari Padmanabhan [1,9,10 ✉], Maxwell Poore[2,10], Peter K. Kim[2,10], Nathan Z. Koocher[3], Vladimir A. Stoica [1], Danilo Puggioni [3], Huaiyu (Hugo) Wang [1], Xiaozhe Shen [4], Alexander H. Reid [4], Mingqiang Gu [3], Maxwell Wetherington[1], Seng Huat Lee [5,6], Richard D. Schaller [7], Zhiqiang Mao [5,6], Aaron M. Lindenberg [4,8], Xijie Wang [4], James M. Rondinelli [3], Richard D. Averitt [2 ✉] & Venkatraman Gopalan[1 ✉]

The emergence of magnetism in quantum materials creates a platform to realize spin-based applications in spintronics, magnetic memory, and quantum information science. A key to unlocking new functionalities in these materials is the discovery of tunable coupling between spins and other microscopic degrees of freedom. We present evidence for interlayer magnetophononic coupling in the layered magnetic topological insulator MnBi$_2$Te$_4$. Employing magneto-Raman spectroscopy, we observe anomalies in phonon scattering intensities across magnetic field-driven phase transitions, despite the absence of discernible static structural changes. This behavior is a consequence of a magnetophononic wave-mixing process that allows for the excitation of zone-boundary phonons that are otherwise 'forbidden' by momentum conservation. Our microscopic model based on density functional theory calculations reveals that this phenomenon can be attributed to phonons modulating the interlayer exchange coupling. Moreover, signatures of magnetophononic coupling are also observed in the time domain through the ultrafast excitation and detection of coherent phonons across magnetic transitions. In light of the intimate connection between magnetism and topology in MnBi$_2$Te$_4$, the magnetophononic coupling represents an important step towards coherent on-demand manipulation of magnetic topological phases.

---

[1] Materials Research Institute and Department of Materials Science & Engineering, Pennsylvania State University, University Park, PA, USA. [2] Department of Physics, University of California San Diego, La Jolla, CA, USA. [3] Department of Materials Science and Engineering, Northwestern University, Evanston, IL, USA. [4] SLAC National Accelerator Laboratory, Menlo Park, CA, USA. [5] 2D Crystal Consortium, Materials Research Institute, Pennsylvania State University, University Park, PA, USA. [6] Department of Physics, Pennsylvania State University,  University Park, PA, USA. [7] Center for Nanoscale Materials, Argonne National Laboratory, Lemont, IL, USA. [8] Department of Materials Science and Engineering, Stanford University, Stanford, CA, USA. [9] Present address: Department of Physics, Harvard University, Cambridge, MA, USA. [10] These authors contributed equally: Hari Padmanabhan, Maxwell Poore, Peter K. Kim. ✉email: hari@psu.edu; raveritt@ucsd.edu; vgopalan@psu.edu

The realization of magnetic order in functional quantum materials creates a rich platform for the exploration of fundamental spin-based phenomena, as exemplified in strongly correlated materials[1], multiferroics[2], and more recently, magnetic topological materials[3]. As such, these materials hold great promise for application in spintronics, magnetic memory, and quantum information technology. A new paradigm has recently emerged with the discovery of atomically thin magnets, derived from layered, quasi-two-dimensional materials[4]. In such materials, magnetic order is characterized by strongly anisotropic exchange interactions, with interlayer exchange coupling that is an order-of-magnitude weaker than the in-plane exchange coupling. The weak interlayer exchange coupling offers a high degree of tunability in the two-dimensional limit, enabling the realization of phenomena such as magnetic switching via electric fields[5] and electrostatic doping[6]. Such tunability could potentially be made even more potent in combination with additional functionalities such as those outlined above. For instance, the $Mn(Bi,Sb)_{2n}Te_{3n+1}$ family of layered antiferromagnets is the first experimental realization of intrinsic magnetic order in topological insulators[7–9]. The interlayer magnetic order is intimately connected to the band topology, with experimental demonstration of switching between quantum anomalous Hall and axion insulator states[10], and realization of a field-driven Weyl semimetal state[11]. In this context, the discovery of new, efficient coupling pathways between the interlayer exchange and other microscopic degrees of freedom would not only add to the rich spectrum of low-dimensional magnetic phenomena, but also potentially unlock pathways for the dynamic manipulation of magnetism and band topology.

In this work, we observe that interlayer magnetic order in $MnBi_2Te_4$ is strongly coupled to phonons, manifesting in the optical excitation of zone-boundary phonons that are otherwise forbidden due to the conservation of momentum. This magnetophononic response is a consequence of a coherent wave-mixing process between the antiferromagnetic order and $A_{1g}$ optical phonons, as determined from equilibrium and time-domain spectroscopy across temperature- and magnetic field-driven phase transitions. Our microscopic model based on first-principles calculations reveals that this phenomenon can be attributed to phonons modulating the interlayer exchange coupling.

## Results

**Spectroscopic evidence of magnetophononic coupling.** $MnBi_2Te_4$ exhibits magnetic order below a temperature of $T_N = 24$ K, with in-plane ferromagnetic coupling, and out-of-plane antiferromagnetic (AFM) coupling[12], as shown in Fig. 1a. With an applied out-of-plane magnetic field, a spin-flop transition occurs at 3.7 T, developing into a fully polarized ferromagnetic-like state (FM) at a critical field of 7.7 T[12]. We first present measurements of the phonon spectra across the magnetic phase transitions in $MnBi_2Te_4$, using magneto-Raman spectroscopy. The full polarized Raman phonon spectra, selection rules, and peak assignments can be found in Supplementary Note 1. Our peak assignment is fully consistent with a previous study[13] that investigated Raman phonons in thin flakes of $MnBi_2Te_4$ as a function of number of layers. Here we focus on two fully symmetric "$A_{1g}$" phonon modes at frequencies of 49 and 113 $cm^{-1}$, labeled $A_{1g}^{(1)}$ and $A_{1g}^{(2)}$ respectively. The phonon eigendisplacements, calculated using density functional theory (DFT) simulations, are shown in Fig. 1b. Representative spectra at 0 T, in the AFM phase at 15 K and the paramagnetic (PM) phase at 35 K, are shown in Fig. 1c, d, respectively. We observe that the $A_{1g}^{(2)}$ mode clearly exhibits an anomalous increase in scattering intensity in the AFM phase, which has not been reported in

previous studies[13]. The temperature-dependence of the $A_{1g}^{(1)}$ mode is discussed in detail in Supplementary Note 2. In the following, we focus on the magnetic field-dependent behavior. At a magnetic field of 9 T, where $MnBi_2Te_4$ is in the fully polarized ferromagnetic (FM) state, the spectral weight of both modes decreases, as shown in Fig. 1e, f. This is highlighted by subtracting the spectrum at 9 T from the spectrum at 0 T and plotting the residual in Fig. 1g, h. In Fig. 1i, j, the residual is plotted as a function of magnetic field $H$, upon subtracting the 9 T spectrum. A clear correlation is observed between the residual scattering intensity of the $A_{1g}$ modes and the critical magnetic fields for the spin-flop and FM transitions, denoted by dashed white lines.

The fractional change in integrated intensity of the $A_{1g}^{(2)}$ mode is plotted as a function of temperature in Fig. 2a (green dots). The integrated intensity follows the AFM order parameter, tracked by the (1 0 5/2) neutron diffraction Bragg peak[14] (purple dots). The gray line is a fit to the power law $I \propto \left(1 - \frac{T}{T_N}\right)^{2\beta}$, with $\beta = 0.35$ as in the reference[14]. Furthermore, a plot of the scattering intensity of the $A_{1g}^{(1)}$ and $A_{1g}^{(2)}$ modes (Fig. 2b) as a function of magnetic field reveals the fractional change in integrated intensities of both modes tracks the AFM order parameter[15] across the spin-flop transition at 3.7 T, and into the fully polarized ferromagnetic state above 7.7 T. The integrated intensities of the $A_{1g}^{(1)}$ and $A_{1g}^{(2)}$ modes increase by fractions of 0.15 and 0.3 respectively, in the AFM phase, as compared to the FM phase at 9 T. Additionally, the fractional increase in the $A_{1g}^{(2)}$ intensity as estimated from the PM to AFM transition and FM to AFM transition in Fig. 2a, b, respectively, are of the same magnitude, pointing to a common origin. Importantly, within the limits of our experimental uncertainty (error bars in plots), we do not observe such large changes in the integrated intensity on any of the other Raman phonons (see Supplementary Note 3 for detailed field-dependent data). Below, we show that the experimentally observed temperature- and field-dependent evolution of scattering intensity is consistent with the excitation of 'forbidden' zone-boundary modes of the $A_{1g}^{(1)}$ and $A_{1g}^{(2)}$ phonon branches.

The AFM order along the out-of-plane direction (crystallographic c-axis) results in a magnetic unit cell that is double the size of the crystallographic unit cell, as shown in Fig. 3a. In contrast, in the high-field FM state (and the paramagnetic state), the magnetic unit cell is identical with the crystallographic unit cell, as in the paramagnetic state. This behavior manifests in the anomalous field-dependent scattering intensity of the $A_{1g}$ modes, which follows the AFM order parameter with the magnetic unit cell doubling resulting in a folding of the phonon Brillouin zone, allowing for the optical detection of zone-boundary phonon modes. DFT simulations of the phonon dispersion along the out-of-plane direction reveal a flat dispersion for the $A_{1g}^{(2)}$ mode, and a small dispersion for the $A_{1g}^{(1)}$, consistent with the weak interlayer van der Waals interaction, and our experimental results, denoted in Fig. 3b using bold circles. This supports our assignment of the anomalous scattering intensity as zone-boundary modes. We also consider and rule out alternative explanations for the observed temperature- and magnetic field-dependent scattering intensity changes, such as resonant Raman effects (see Supplementary Note 5) and possible magnon resonances overlapping with the considered phonons (see Supplementary Note 6).

Magnetic unit cell doubling resulting in the activation of zone-boundary phonons is unexpected given the absence of a structural phase transition. Refinement based on neutron diffraction at 10 and 100 K shows no structural unit cell doubling across the AFM transition, and no changes to the unit cell coordinates to within $10^{-3}$ of the lattice parameters[14]. The negligible change in the

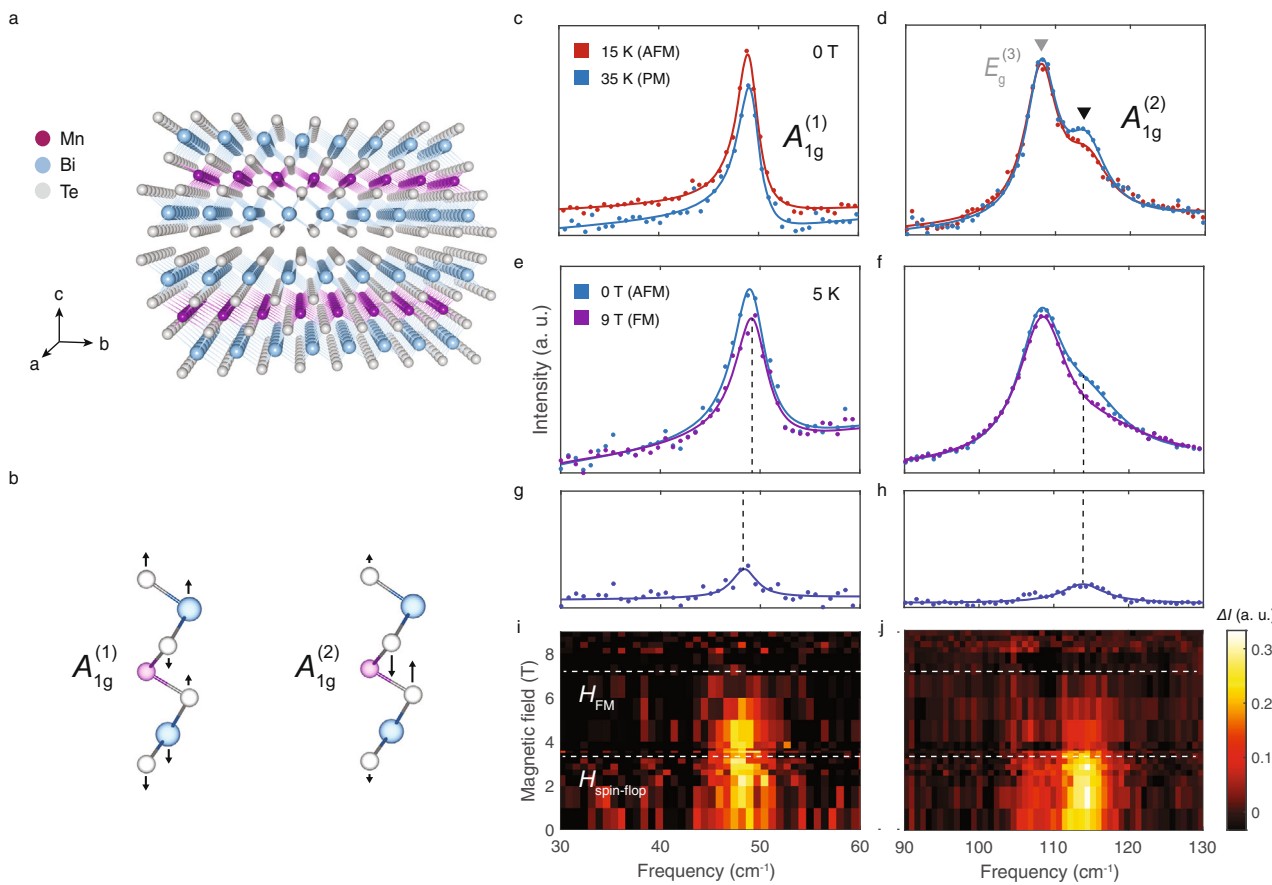

**Fig. 1 Phonon anomalies across magnetic phase transitions in MnBi$_2$Te$_4$. a** Crystal structure of MnBi$_2$Te$_4$. **b** Eigendisplacements of the $A_{1g}^{(1)}$ and $A_{1g}^{(2)}$ modes, with arrows denoting displacement of ions. **c**, **d** Raman spectra of $A_{1g}^{(1)}$ (c) and $A_{1g}^{(2)}$ (d) modes in the paramagnetic (PM) and antiferromagnetic (AFM) phases at 0 T, shown in red and blue respectively. **e**, **f** Raman spectra of $A_{1g}^{(1)}$ (e) and $A_{1g}^{(2)}$ (f) modes in the AFM and ferromagnetic (FM) phases at 5 K, shown in blue, and purple respectively. (**g**, **h**) The difference between spectra in the AFM and FM phases. **i**, **j** Contour plots of the difference upon subtracting the 9 T spectrum, as a function of magnetic field. The dotted lines denote the FM and spin-flop critical fields.

spectra of other Raman phonons in MnBi$_2$Te$_4$ is also consistent with the absence of a structural transition of any kind, and points instead to a mechanism that is mode-dependent.

**Microscopic model of magnetophononic wave-mixing.** In general, zone-boundary modes are optically inactive or "forbidden" due to the conservation of crystal momentum. Photons in the visible part of the spectrum have negligible momentum in comparison with the crystal Brillouin zone, and thus momentum conservation dictates that only zero momentum (i.e., zone-center) excitations can be generated and detected in first-order scattering processes. This is shown schematically for Raman scattering in Fig. 3c. This selection rule can be overcome in the presence of other finite-momentum waves in the crystal, as observed for instance in the case of structural distortions that double the crystallographic unit cell[16–18]. However, as noted above, MnBi$_2$Te$_4$ does not exhibit any structural transition. Instead, we propose that the crystal momentum is provided by the AFM order, via a magnetophononic wave-mixing process. This is shown schematically in Fig. 3c, where the AFM crystal momentum $q_{AFM} = 2\pi/2c$ interacts with the phonon crystal momentum, allowing for the excitation of zone-boundary ($q = \pi/c$) phonons.

Magnetophononic wave-mixing requires a sufficiently strong scattering cross-section to be observable. This scattering cross-section can typically be written in terms of an interaction term in the free energy. For example, the Raman scattering process is due to

a coupling of the incident ($E^i$) and reflected ($E^r$) electric fields to a distortion $u$ along a phonon normal mode, via the susceptibility $\chi_e$ (i.e., $F = \left(\frac{d\chi_e}{du} u\right) E^i E^r$). In the case of a finite-momentum structural distortion, phonons couple to the structural distortion through elastic interactions. Analogously, in our model of magnetophononic wave-mixing, phonons couple to the AFM order by modulating the interlayer exchange interaction $J^\perp$. The corresponding interaction term in the free energy can be obtained by first writing down a Heisenberg-like Hamiltonian for the spin energy, $H = \sum_{ij} J_{ij} S_i \cdot S_j$, where $J_{ij}$ is the exchange coupling between spins at sites $i$ and $j$. Since the coupling is to an out-of-plane antiferromagnetic spin wave, we focus on the interlayer (out-of-plane) exchange coupling $J^\perp$ (only nearest-neighbor interlayer interactions are considered). If a phonon modulates the interlayer exchange interaction, the perturbed exchange coupling $J^{\perp\prime}$ can be written as

$$J^{\perp\prime} = J^\perp + \frac{dJ^\perp}{du} u + \dots \quad (1)$$

Equation 1 is a special case of what is broadly referred to in the literature as "spin–phonon coupling" (see Supplementary Note 4 for the interpretation of higher-order terms in terms of phonon frequency renormalization). Based on this, the free energy term that couples the antiferromagnetic spin wave to the phonon is, to

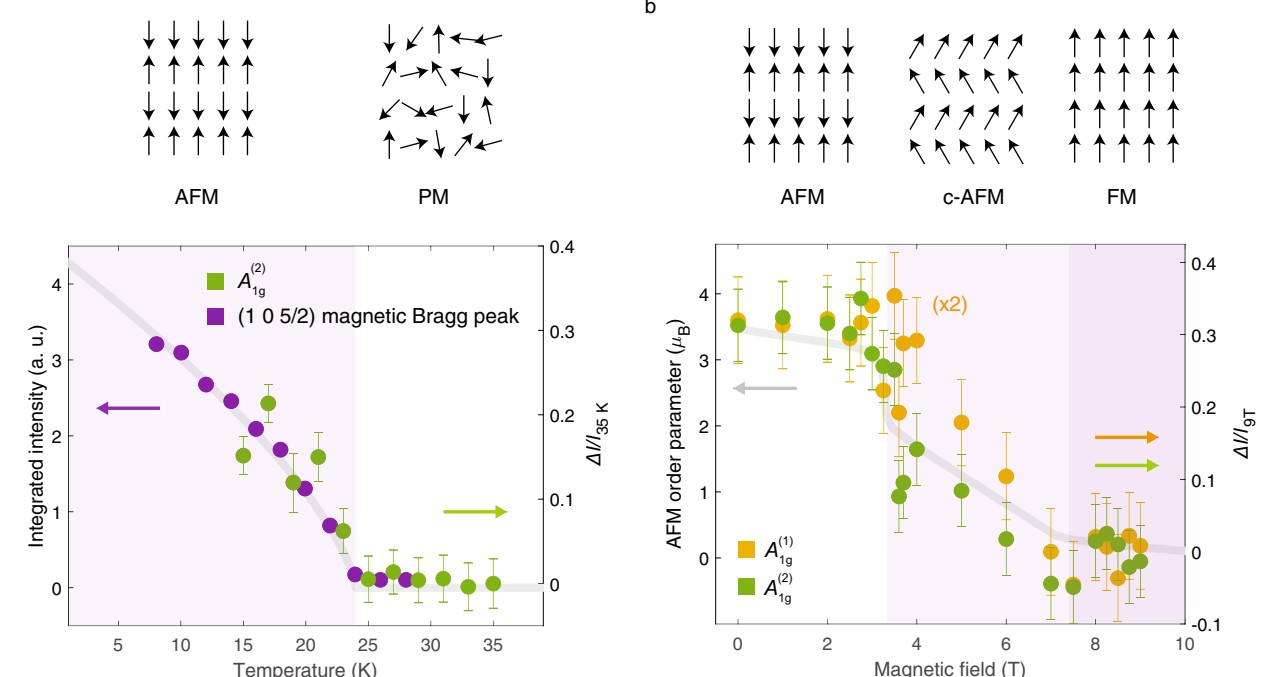

**Fig. 2 Phonon intensities track the antiferromagnetic order parameter. a** Temperature-dependent fractional change in integrated intensity, $\Delta I/I_{35K}$, of the $A_{1g}^{(2)}$ mode, overlayed on integrated intensity of the (1 0 5/2) neutron diffraction peak from reference[14]. The gray line is a fit to $A\left(1 - T/T_N\right)^{2\beta}$, with $\beta = 0.35$, $T_N = 24$ K. **b** The field-dependent fractional change in integrated intensity, $\Delta I/I_{9T}$, of the $A_{1g}^{(1)}$ and $A_{1g}^{(2)}$ modes. The gray line is the AFM order parameter, given by $M - 4.5\,\mu_B$, where $M$ is the magnetization measured by magnetometry from reference[15]. Error bars are standard deviations in fit values.

first order,

$$F = \left(\frac{dJ^\perp}{du}u\right)\sum_i S_i S_{i+1} \qquad (2)$$

where $i$ and $i+1$ correspond to nearest-neighbor spin pairs in the out-of-plane direction. It is clear that the magnitude of this coupling directly depends on $\frac{dJ^\perp}{du}$. In other words, a magnetophononic wave-mixing is possible only when the phonon mode under consideration sufficiently modulates the interlayer exchange coupling.

A microscopic basis for this model can be obtained using DFT simulations. We simulate the modulation of the interlayer exchange coupling $J^\perp$ by the six Raman phonons of MnBi$_2$Te$_4$, which include three $A_{1g}$ modes (pure out-of-plane eigendisplacements), and three $E_g$ modes (pure in-plane eigendisplacements, see Supplementary Fig. 1b for eigendisplacements). The results, shown in Fig. 3d, indicate a striking dichotomy between the out-of-plane $A_{1g}$ modes and in-plane $E_g$ modes. The $A_{1g}$ modes exhibit an order-of-magnitude larger modulation of $J^\perp$ than the $E_g$ modes. Furthermore, the $A_{1g}^{(2)}$ mode has by far the largest influence on $J^\perp$, consistent with our experimental observation of zone-boundary scattering intensity. A quantitative comparison of this model with our experimental results is possible. This is accomplished by defining an experimental magnetophononic scattering cross-section $\sigma$, as the ratio of the integrated intensity of the zone-boundary mode (i. e. the residual spectra in Fig. 1g, h) to that of the zone-center mode (spectra at 9 T in Fig. 1e, f). The scattering cross-section is compared to the calculated interaction term, $\left|\frac{dJ^\perp}{du}\right|$. The plotted results in Fig. 3e show a good agreement between theory and the experiment. In particular, the model reproduces the experimental observation of the $A_{1g}^{(2)}$ mode exhibiting the largest zone-boundary scattering intensity. We note that no signature of a zone-boundary mode was observed in the $A_{1g}^{(3)}$ branch within the experimental

uncertainty (see Supplementary Note 3). Finally, also in agreement with the theoretical prediction, no $E_g$ zone-boundary modes were experimentally observed, i.e., $\sigma = 0$ for all $E_g$ modes, within the experimental uncertainty (see Supplementary Note 3).

The theoretical results outlined above can be rationalized in terms of microscopic interlayer exchange pathways. In general, the exchange coupling across a van der Waals (vdW) gap is understood to be the result of a process named "super-superexchange" (SSE)[19]. In SSE, given that the interlayer exchange interaction is usually much weaker than the intralayer exchange interaction, the two can be effectively decoupled. The individual quasi-two-dimensional layers are treated as macroscopic magnetic moments established by the intralayer superexchange (shown in pink in Fig. 3f), which couple across the vdW gap via the weaker interlayer exchange (shown in blue in Fig. 3f). As in any exchange process, geometrical parameters that influence the relevant hopping integrals play a major role. In superexchange, the angle between magnetic ions and its ligands mediates the superexchange, in this case the Mn–Te–Mn bond angle $\theta$ shown in Fig. 3f. These structural superexchange interactions are further controlled by orbital hybridization with cationic Bi $p$ states tuned by the nearest-neighbor ions across the vdW gap[20], in this case, determined by the Te–Te distance $\Delta$ shown in Fig. 3f, to stabilize the FM interlayer coupling in MnBi$_2$Te$_4$.

We first note that $A_{1g}$ modes in MnBi$_2$Te$_4$ modulate $\Delta$, whereas $E_g$ modes do not, an observation that accounts for the dichotomy of their respective influence on $J^\perp$. Of the $A_{1g}$ modes, examining the eigenvectors in Fig. 1b and Supplementary Fig. 1b, A1g$^{(2)}$ exhibits the largest modulation of the Mn–Te–Mn bond angle $\theta$. The modulation of $\theta$ by the $A_{1g}^{(2)}$ mode is a factor of 2 larger than by $A_{1g}^{(1)}$, which in turn is a factor of 5 larger than by $A_{1g}^{(3)}$. This rationalizes the trend seen in the calculated $\frac{dJ^\perp}{du}$ in terms of the SSE pathways.

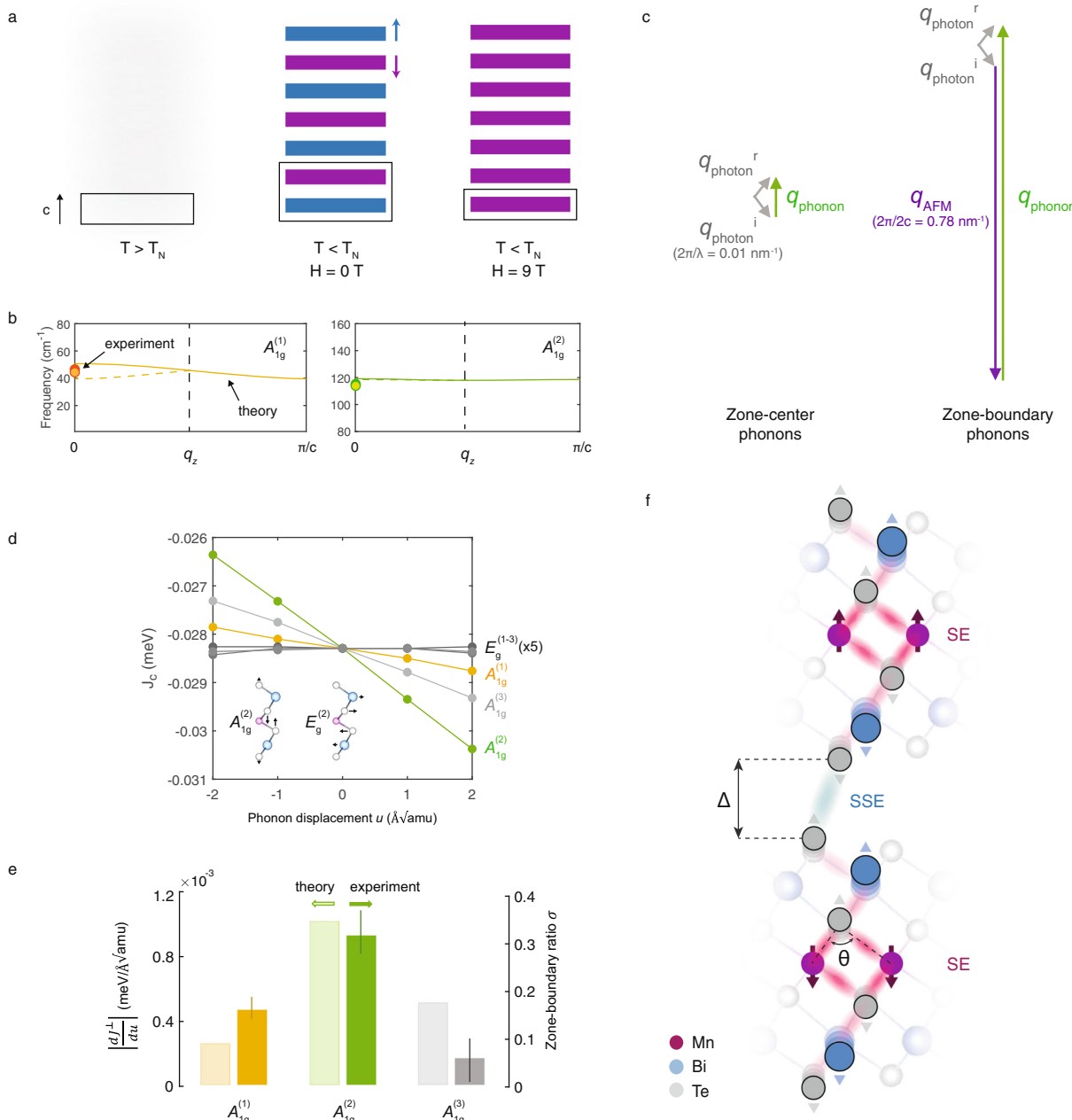

**Fig. 3 Magnetophononic wave-mixing. a** Schematic of layered magnetic ordering in MnBi$_2$Te$_4$, with blue and purple denoting opposite in-plane spin orientations, and gray denoting disordered spins. The antiferromagnetic (AFM) wavevector is shown schematically, labeled "q$_{AFM}$". **b** The dispersion relations of the $A_{1g}^{(1)}$ and $A_{1g}^{(2)}$ modes along the c-axis, calculated using density functional theory. The experimental zone-center and zone-boundary phonon frequencies are denoted using colored and empty circles respectively. **c** Schematic of wave-mixing for zone-center and zone-boundary modes. The wavevectors of the photon (i = incident, r = reflected), phonon, and AFM spin-wave are shown using gray, green, and purple arrows (not drawn to scale). **d** Modulation of the interlayer exchange coupling $J^\perp$ by Raman phonons. Inset shows the eigendisplacements of two representative phonons. **e** Comparison between the calculated magnetophononic scattering cross-section $\left|\frac{dJ^\perp}{du}\right|$ and the experimental zone-boundary ratio $\sigma$ (see text for definition). Error bars are standard deviations in fit values. **f** Schematic of superexchange (SE) and super-superexchange (SSE), with Δ denoting the interlayer distance, θ denoting the Mn–Te–Mn bond angle, and pink and blue clouds denoting SE and SSE pathways, respectively.

**Time-domain signatures of magnetophononic coupling.**
Finally, we investigate magnetophononic coupling by direct measurement of phonons in the time domain. To do this, we carry out "pump-probe" experiments to generate and detect coherent optical phonons as a function of magnetic field (see schematic in Fig. 4a). Excitation with ultrafast optical pump pulses (1.55 eV, 50 fs) results in the generation of coherent phonon oscillations. A second, time-delayed probe pulse (1.2 eV, 50 fs) measures pump-induced changes in the transient reflectivity (ΔR/R). The transient

reflectivity is sensitive to changes in carrier density and coherent phonons. These measurements are carried out at 2 K, as a function of magnetic field from 0 to 6.4 T, across the spin-flop transition. The transient reflectivity, shown in Fig. 4b, exhibits an initial sub-picosecond dip, followed by a slow relaxation. Overlayed on this are multiple distinct coherent oscillation components that (as described below), correspond to the $A_{1g}^{(1)}$ and $A_{1g}^{(2)}$ phonons. We normalize the pump-probe reflectivity traces with respect to their maximum amplitudes, to account for field-

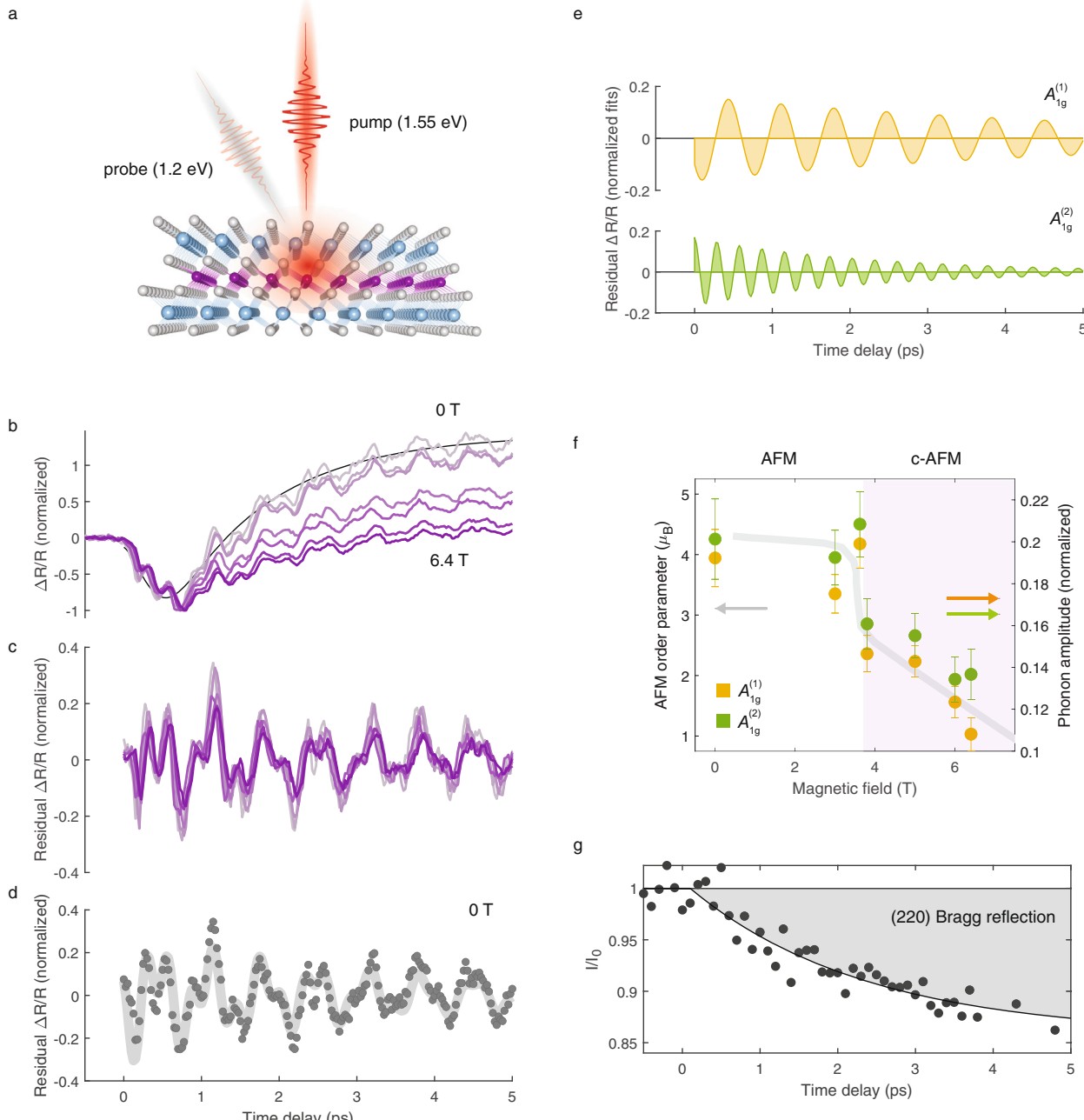

**Fig. 4 Ultrafast signatures of magnetophononic coupling. a** Schematic of pump-probe measurement. **b** Pump-induced changes in the transient reflectivity ($\Delta R/R$) as a function of time delay at various magnetic fields, normalized to the maximum amplitude. The black line is a representative biexponential fit to the 0 T data. **c** The residual $\Delta R/R$ upon subtracting a biexponential fit. **d** Residual $\Delta R/R$ at 0 T, with black dots denoting experimental datapoints and the gray line denoting the fit to the sum of two decaying sinusoidal functions. **e** Individual decaying sinusoidal components obtained from the fit in (**b**), corresponding to the $A_{1g}^{(1)}$ (top) and $A_{1g}^{(2)}$ (bottom) phonons, respectively. **f** Initial amplitude of the coherent $A_{1g}^{(1)}$ and $A_{1g}^{(2)}$ phonons, obtained from fit result in (**d**). The gray line is the antiferromagnetic order parameter from reference[15]. **g** Measured transient electron diffraction intensity of the (2 2 0) Bragg peak, with black dots denoting experimental datapoints, and the black line denoting the fit to an exponential decay function. Error bars are standard deviations in fit values.

dependent variation in the absorbed fluence, and thus photo-carrier density, which can influence coherent phonon amplitudes (see Supplementary Note 7 and Supplementary Fig. 8 for detailed discussion). Upon subtracting biexponential fits (black line fit to 0 T data in Fig. 4b shown as a representative example), we observe that the normalized phonon oscillation amplitudes in the residual $\Delta R/R$ in Fig. 4c visibly decrease with increasing magnetic field, much like the phonon spectral weights measured using Raman spectroscopy.

The individual oscillatory components are obtained by fitting the residual $\Delta R/R$ to the sum of two exponentially decaying sinusoidal functions (see Methods) as shown for the representative 0 T data in Fig. 4d The individual sinusoidal functions, shown in Fig. 4e, are readily identified as the $A_{1g}^{(1)}$ and $A_{1g}^{(2)}$ modes at 1.47 THz (49 cm$^{-1}$) and 3.44 THz (115 cm$^{-1}$), respectively. Plotting the amplitudes of the two coherent phonon modes as a function of magnetic field in Fig. 4e, it is clear that both modes track the AFM order parameter denoted by the solid gray line, in

striking similarity to the field-dependent change in the Raman scattering intensities.

The detection of coherent phonons in pump-probe experiments occurs through a process that is identical to spontaneous Raman scattering[21,22]. The generation of coherent phonons can also be described within a Raman formalism, with the real and imaginary parts of the Raman tensor responsible for phonon excitation in transparent and absorbing materials, respectively[21]. The similarity of the magnetic-field-dependent coherent phonon amplitudes in Fig. 4f to the static Raman scattering intensities in Fig. 2b thus suggests that these are a consequence of the same mechanism, namely the excitation of zone-boundary phonons via the crystal momentum associated with the antiferromagnetic order.

For resonant excitation of $MnBi_2Te_4$ with 1.55 eV pulses, phonon excitation through the imaginary part of the Raman tensor may be physically thought of in terms of a "displacive" excitation[23], where the ultrafast excitation of carriers by the pump pulse shifts the quasi-equilibrium coordinates of the lattice in a spatially and temporally coherent manner, generating coherent phonons. Within this picture, magnetophononic zone-folding as described in the previous section would allow for the generation of both zone-center as well as nominally zone-boundary $A_{1g}$ modes. Additionally, the electronic excitation that shifts the quasi-equilibrium coordinates may itself have a $q_z = \pi/c$ component owing to the contrast in spin-split electronic bands in alternating layers, acting as a direct driving force for the generation of zone-boundary phonons. Unfortunately, the small frequency splitting of the $A_{1g}$ modes precludes the explicit resolution of zone-boundary phonons in the time domain. Nonetheless, it is clear from Fig. 4f that the coherent phonons track the AFM order parameter in accord with the magnetophononic wave-mixing proposed here.

We note that in general, phonons in time-domain measurements are expected to exhibit qualitative deviations from steady-state spectroscopy, owing to the nonequilibrium nature of the former. While the ultrafast carrier excitation in displacive phonon excitation is itself a manifestly nonequilibrium process, additional deviations may emerge from nonequilibrium phonon interactions. We directly measure the timescale of phonon equilibration using ultrafast electron diffraction (see Methods). Here, pump-induced changes in the root-mean-square displacements $\langle u^2 \rangle$ of ions through carrier-lattice and lattice thermalization appear in the transient intensity of Bragg peaks through the Debye-Waller effect (see Supplementary Note 8). These measurements require an order-of-magnitude higher pump excitation fluence than the optical pump-probe measurements (see Methods) in order to produce a discernible signal. Regardless, these high fluence measurements set a lower bound for the phonon thermalization time, as discussed in Supplementary Note 8. As a representative sample, we show in Fig. 4g, the transient intensity of the (2 2 0) Bragg peak, with the evolution of the peak intensity fit to an exponential decay (black line). The results indicate that phonon populations indeed remain in a nonequilibrium state through the entire time delay range considered. It is noteworthy that clear signatures of magnetophononic coupling are observed even under such nonequilibrium conditions. Finally, we mention that there may possibly be additional contributions to the coherent phonon amplitudes from magnetodielectric effects which are not explicitly accounted for here. We discuss the possible contributions to coherent phonon amplitudes due to such an effect Supplementary Note 7.

## Discussion
We have demonstrated that optically "forbidden" zone-boundary phonons are observed due to magnetophononic wave-mixing in

$MnBi_2Te_4$. While it is uncommon for purely magnetic unit cell doubling to give rise to phonon zone-folding effects, such signatures were first observed in transition metal dihalides[24]. These observations were rationalized in terms of phenomenological models of electron-phonon coupling that took into consideration phonon modulation of the spin–orbit coupling and exchange interactions[24,25]. Our model instead considers the scattering cross-section between the AFM order and phonons, arriving at qualitatively similar conclusions. Importantly, our work provides a description of such a model using first-principles theory. The excellent agreement between the theory and experimental results not only validates the model, but also provides a microscopic basis for the observed phenomena in terms of SSE inter-layer exchange pathways. Our work may also help rationalize similar phenomena recently reported[26,27] in other quasi-two-dimensional magnets such as $CrI_3$ and $FePS_3$.

Our discovery is especially of significance in light of the critical role played by tunable interlayer exchange interactions in layered magnetic materials. For instance, in $MnBi_2Te_4$, the interlayer magnetic ordering can drive topological phase transitions between quantum anomalous Hall and axion insulator states. Our work unlocks the possibility of controlling the interlayer magnetic ordering in $MnBi_2Te_4$ by exploiting the strong coupling of $A_{1g}$ phonons to $J^\perp$. A promising route towards the ultrafast control of magnetism in $MnBi_2Te_4$ is the use of resonant THz excitation to drive large amplitude distortions along $A_{1g}$ modes, as opposed to employing carrier-based mechanisms (such as displacive excitation) that suffer from ultrafast heating effects, which limit the amplitude of coherent phonons. This may be through anharmonic coupling to Raman-active modes[28], or alternatively through sum-frequency ionic Raman scattering[29]. Such mechanisms based on resonant coupling have been used to drive ultrafast light-induced magnetic oscillations and phase transitions, as experimentally demonstrated in other materials[30–36]. Experimental studies[37] on $Bi_2Se_3$, a material closely related to $MnBi_2Te_4$, have demonstrated the feasibility of ionic Raman scattering as a way to drive large amplitude oscillations along Raman-active modes. Recent theoretical work[38] has outlined an approach based on anharmonic phonon interactions in $MnBi_2Te_4$. In particular, it was shown that resonant excitation of an IR-active $A_{2u}$ phonon (at a frequency of 156 $cm^{-1}$ = 4.7 THz) could drive large amplitude oscillations, which via anharmonic coupling, would drive a unidirectional distortion along Raman-active $A_{1g}$ modes such as the ones identified in the present work. It was predicted that such an approach could be used to drive an AFM to FM transition concurrent with a topological phase transition, using experimentally accessible ultrafast modalities. The magnetophononic wave-mixing in the present work provides an experimental foundation for such approaches and a path toward achieving ultrafast light-induced topological phase transitions.

## Methods
**Crystal growth and characterization**. Single crystals of $MnBi_2Te_4$ were grown using a self-flux method[11]. Mixtures of 99.95% purity manganese powder, 99.999% bismuth shot, and 99.9999+% tellurium ingot with a molar ratio Mn:Bi:Te = 1:10:16 were loaded into an aluminum crucible and sealed in evacuated quartz tubes. The mixture is heated upto 1173 K for 12 h and slowly cooled down at the rate of 1.5 K/h to 863 K. This is followed by centrifugation to remove excess flux. The phase and crystallinity of the single crystals were checked by X-ray diffraction. The antiferromagnetic order with the Néel temperature of 24 K was confirmed using SQUID magnetometry.

**Raman spectroscopy measurements**. Temperature-dependent Raman spectra were collected using a Horiba LabRam HR Evolution with a freespace Olympus BX51 confocal microscope. A 632.8 nm linearly polarized HeNe laser beam was focused at normal incidence using a LWD 50× objective with a numerical aperture of 0.5, with the confocal hole set to 100 μm. A Si back-illuminated deep depleted

array detector and an ultra-low-frequency volume Bragg filter were used to collect the spectra, dispersed by a grating (1800 gr/mm) with an 800 mm focal length spectrometer. The system was interfaced with an Oxford continuous-flow cryostat for low-temperature measurements, using liquid helium as the cryogen.

Field-dependent magneto-Raman spectra were collected using a home-built Raman spectrometer. A 632.8 nm linearly polarized HeNe laser beam was focused at normal incidence using a LWD 50x objective with a numerical aperture of 0.82. A Si back-illuminated deep depleted array detector and a set of ultra-low-frequency volume Bragg filters were used to collect the spectra, dispersed by a grating (1800 gr/mm) with a 300 mm focal length spectrometer. The system was interfaced with an Attocube AttoDRY 2100 closed-cycle cryostat for low-temperature, high magnetic-field measurements, using liquid Helium as the cryogen. The field-induced Faraday rotation in the objective was calibrated and corrected using a half-waveplate.

The laser power was maintained below 50 μW in all measurements, to minimize laser heating and maintain the power well below the damage threshold. Laser heating was calibrated by measuring Raman phonon peak shifts as a function of and using thermal conductivity values from reference[14]. Polarized spectra were obtained using a half-waveplate to rotate the polarization of the incident beam, with a fixed analyzer.

After peak assignment using polarization analysis, temperature- and field-dependent spectra were collected without a polarizer, to maximize signal throughput. Spectra were averaged over 60 and 120 min in the case of temperature-dependent and field-dependent measurements respectively, with a temperature stability of ±0.1 K. Any subtle drift in the spectrometer (<0.15 cm$^{-1}$) over the temperature-dependent studies was corrected using the HeNe line at 632.8 nm.

The $A_{1g}^{(1)}$ peak was fit using an inverse Fano lineshape in combination with a linear background. Its lineshape is given by the expression $(\omega) = \frac{(q\Gamma - (\omega - \omega_0))^2}{\Gamma^2 + (\omega - \omega_0)^2}$, where $I$ is the scattering intensity, $\omega$ is the energy, $\omega_0$ and $\Gamma$ are the resonant energy and linewidth of the excitation respectively, and $1/q$ is a measure of the peak asymmetry. The $E_g^{(2)}$, and $A_{1g}^{(3)}$ peaks were fit with a standard Gaussian lineshape, and the $E_{1g}^{(3)}$ and $A_{1g}^{(2)}$ peaks were fit with a standard Lorentzian lineshape.

A nonlinear least-squares fitting procedure was used. To ensure robustness of the temperature-dependent fits, the same initial fit values and constraints were used for each set of temperature-dependent and field-dependent spectra.

### Magnetic field-dependent ultrafast optical spectroscopy.
Ultrafast optical pump-probe measurements were carried out using a 1040 nm 200 kHz Spectra-Physics Spirit Yb-based hybrid-fiber laser coupled to a noncollinear optical parametric amplifier. The amplifier produces <50 fs pulses centered at 800 nm (1.55 eV), which is used as the pump beam. The 1040 nm (1.2 eV) output is converted to white light, centered at 1025 nm with a FWHM of 20 nm, by focusing it inside a YAG (Yttrium Aluminum Garnet) crystal. The white light is subsequently compressed to ~50 fs pulses using a prism compressor pair and is used as the probe beam. The pump and the probe beams are aligned to propagate along the [001] axis of the crystal, at near normal incidence.

The samples were placed in a magneto-optical closed-cycle cryostat (Quantum Design OptiCool). Pump-probe measurements were carried out as a function of magnetic field applied normal to the sample surface (along the [001] direction). The sample temperature was fixed at 2 K. A pump fluence of ~100 μJ/cm$^2$ was used in order to generate sufficiently large coherent phonon oscillations, while keeping the transient heating to a minimal amount, to ensure we avoid melting of the magnetic order.

### Ultrafast electron diffraction measurements.
Ultrafast electron diffraction measurements were carried out at the MeV-UED beamline at the SLAC National Accelerator Laboratory. The principle and other technical details of the experimental setup are outlined elsewhere[39]. A 60-fs laser pulse with a photon energy of 1.55 eV and fluence of 7 mJ/cm$^2$ were used to excite the sample. A higher pump fluence was required than in the optical pump-probe measurements, in order to produce a sufficiently large pump-induced change in diffraction intensities. Fluence-dependent damage studies revealed no signs of laser-induced damage, and the measurements were repeatable over thousands of cycles. Femtosecond electron bunches of ~100 fs pulsewidth and 3.7 MeV kinetic energy were used to measure pump-induced changes in electron diffraction intensities.

Measurements were carried out on flakes with an average thickness of around 100 nm, exfoliated from a single crystal of $MnBi_2Te_4$ and transferred onto an amorphous $Si_3N_4$ membrane using an ex-situ transfer stage. The flakes were protected with an additional layer of amorphous $Si_3N_4$ to prevent degradation. The spot sizes of the pump and probe beams were $464 \times 694$ μm and ~70 μm, respectively, and the measurements were carried out at 30 K.

The ultrafast electron diffraction intensities were obtained by averaging over several scans, normalizing individual diffraction images to account for electron beam intensity fluctuation. Individual diffraction peaks were fit to a two-dimensional Gaussian function, and then averaged over symmetry-related peaks based on the $R\text{-}3m$ space group of $MnBi_2Te_4$.

### Pump-probe data analysis.
The time-resolved reflectivity traces were first fitted to a product of an error function and a biexponential decay function. The error function models the excitation of photo-carriers and instrumental temporal resolution, and the exponential decay is an approximation for the sum of various unknown processes occurring over the measured time delay, including electron-electron and electron-phonon thermalization. The functional form is:

$$\left(1 + \text{erf}\left(\frac{t}{\tau_r}\right)\right) \times \left(A_1 \exp\left(-\frac{t}{\tau_1}\right) + A_2 \exp\left(-\frac{t}{\tau_2}\right) + C\right)$$

where $t$ is the time delay, $\tau_{el}$ is the rise time for the excitation of photo-carriers, $\tau_1$ and $\tau_2$ are the time constants of exponential decay, and $A_1$, $A_2$, and $C$ are constants.

Upon subtracting the biexponential decay, the residual traces were fit to the sum of two decaying sinusoidal functions. The functional form is:

$$A_1 \sin(2\pi f_1 t + \phi_1) \exp\left(-\frac{t}{\tau_{d1}}\right) + A_2 \sin(2\pi f_2 t + \phi_2) \exp\left(-\frac{t}{\tau_{d2}}\right)$$

where $t$ is the time delay, $f_1$ and $f_2$ are the frequencies of the sinusoidal functionals, corresponding to the $A_{1g}^{(1)}$ and $A_{!g}^{(2)}$ phonons, $\phi_1$ and $\phi_2$ are the phases, and $\tau_{d1}$ and $\tau_{d2}$ are the time constants of exponential decay of the oscillations. The initial amplitudes $A_1$ and $A_2$ are plotted in Fig. 4d.

The ultrafast electron diffraction intensities were fit to an exponential decay function of the form:

$$A_1 \exp\left(-\frac{t}{\tau_l}\right) + C$$

where $t$ is the time delay, $\tau_l$ is the time constant, and $A_1$ and $C$ are constants.

### Electronic structure and phonon calculations.
DFT calculations were carried out using the Vienna Ab Initio Simulation Package (VASP)[40–44] with the PBE exchange correlation functional[45] and van der Waals correction via the DFT-D3[46,47] method with Becke-Jonson damping. A Hubbard $U$ was also added to the Mn (4 eV) using Dudarev's[48] approach. A non-primitive cell containing two Mn atoms was used to obtain the equilibrium geometry of the system with AFM-A magnetic structure. Γ-point phonons were obtained with the finite displacement method on a $1 \times 1 \times 1$ "supercell" using the PHONOPY software package[49] and VASP. An energy cutoff of 300 eV was used for all calculations. A $4 \times 4 \times 4$ Γ-centered $k$-point mesh was used for equilibrium relaxations and phonon calculations. The general energy convergence threshold was $1 \times 10^{-8}$ eV and the force convergence threshold for relaxation was $1 \times 10^{-5}$ eV/Å. When including SOC in the magnetic parameter calculations, however, the energy convergence threshold was $1 \times 10^{-6}$ eV. Gaussian smearing with a 0.02 eV width was also used in all relaxation and single-point energy calculations. Density of states calculations employed the tetrahedron method. The metallic state was modeled by electron doping the unit cells with 0.1 electron/Mn atom. Supercells for magnetic exchange calculations were generated using VESTA[50].

### Exchange coupling constants calculations.
Magnetic exchange parameters were obtained by considering a model spin Hamiltonian of the form $H = -\sum_{\langle ij \rangle} J_{ij} S_i \cdot S_j$, where $J_{ij}$ includes intralayer exchange parameters $J_1$ and $J_2$ and interlayer exchange parameter $J^\perp$. A $\sqrt{2} \times \sqrt{2} \times 1$ supercell of the conventional cell was used get the intralayer exchange parameters, while a $1 \times 1 \times 2$ supercell of the primitive cell was used to get the interlayer exchange parameter. Γ-centered $k$-point meshes of $4 \times 4 \times 1$ and $4 \times 4 \times 4$ were used in the respective calculations.

For the intralayer exchange parameters, one FM and two AFM configurations (stripe and up-up-down-down) were used. The spin exchange energy equations in terms of magnetic exchange parameters for structures of $R\bar{3}m$ symmetry are as follows:

$$E_{FM} = 3E_{NM} - 60J_1 S_i \cdot S_j - 60J_2 S_i \cdot S_j$$

$$E_{AFM1} = 3E_{NM} + 12J_1 S_i \cdot S_j + 12J_2 S_i \cdot S_j$$

$$E_{AFM2} = 3E_{NM} + 12J_1 S_i \cdot S_j - 12J_2 S_i \cdot S_j$$

For the interlayer exchange parameter, one FM and one AFM configuration were used.

$$E_{FM} = E_{NM} - 6J^\perp S_i \cdot S_j$$

$$E_{AFM-A} = E_{NM} + 6J^\perp S_i \cdot S_j$$

The calculated values were multiplied by $S^2$ to obtain the exchange coupling in meV, assuming the spin of the local moment is $S = 5/2$.

### Data availability
Supplementary Information is available for this paper. All the data generated in this study have been deposited in the Figshare database at https://doi.org/10.6084/m9.figshare.19102934.v1.

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

## Acknowledgements

H.P., V.A.S., H.W., P.K., M.P., N.Z.K., A.M.L., R.A., J.M.R., and V.G. acknowledge support from the DOE-BES grant DE-SC0012375. H.P. acknowledges partial support from the DOE Computational Materials program, DE-SC0020145. Support for crystal growth and characterization was provided by the National Science Foundation through the Penn State 2D Crystal Consortium-Materials Innovation Platform (2DCC-MIP) under NSF cooperative agreement DMR-1539916 and DMR-2039351. D.P. was supported by the Army Research Office (ARO) under grant no. W911NF-15-1-0017. SLAC MeV-UED is supported in part by the DOE-BES SUF Division Accelerator & Detector R&D program, the LCLS Facility, and SLAC under Contract Nos. DE-AC02–05-CH11231 and DE-AC02–76SF00515. Use of the Center for Nanoscale Materials, a DOE Office of Science User Facility, was supported by the U.S. Department of Energy, Office of Science, Office of Basic Energy Sciences, under Contract No. DE-AC02-06CH11357. Zero-field Raman measurements were performed in the Materials Characterization Laboratory within the Materials Research Institute at Penn State.

## Author contributions

H.P. and V.G. conceived the project. Raman spectroscopy measurements and analysis were carried out by H.P., H.W., M.W., and V.G. Pump-probe reflectivity was carried out by P.K., M.P., H.P., R.S., and R.A., and the results analyzed by H.P., P.K., M.P., and V.A.S. DFT calculations were done by N.Z.K., D.P., M.G., and J.M.R. Ultrafast electron diffraction was carried out by H.P., V.A.S., H.W., X.S., A.H.R., A.M.L., and X.W., and the results were analyzed by H.P. Crystal growth and characterization were done by S.H.L. and Z.Q.M. The paper was written by H.P. with inputs from all authors.

## Competing interests

The authors declare no competing interests.

**Additional information**

