## [Peer Review File · Nature Communications]

Interlayer magnetophononic coupling in MnBi₂Te₄REVIEWER COMMENTS

Reviewer #1 (Remarks to the Author):

The manuscript H. Pardmanabhan et al. reports on demonstrating a coherent coupling between displacively excited optical phonon mode and magnetization in the topological insulator MnBi₂Te₄ as revealed by time-resolved two-color pump-probe experiment. These conclusions are based on observing a modulation of the magneto-optical Kerr signal by coherent oscillations corresponding to the Ag Raman-active phonon mode. This is further corroborated by static Raman spectroscopy data showing a correlation between the onset of the magnetic order and shift in the frequency of the Ag mode as well as by DFT calculations.

Although I find the Raman spectroscopy data convincing and theory somewhat interesting, the interpretation of the time-resolved experiment, which is the main subject of the study, raises serious concerns. Below I provide a detailed list of major and minor comments to the current version of the manuscript.

Major comments:

1. The authors rely on fact that the transient MOKE signal $\Delta\theta$ is proportional to the magnetization ΔM , via the polar Kerr effect. However, it is commonly known that one of the proportionality factors relies on the intensity of the incident/reflected light R , such that:

$$\Delta\theta \sim \Delta R * M + R * \Delta M \quad (1)$$

implying that modulation of the light intensity ΔR can contribute to the transient Kerr rotation. That is the reason why the polarization rotation can be in principle be also subjected to a modulation due to phonon modes. It is known that even in the absence of magnetization the time-resolved polarization rotation can be modulated by the coherent acoustic and optical phonons (e.g. J. Appl. Phys. 114, 093513 (2013)) due to the transient birefringence.

The authors make a fair attempt to clarify whether the oscillations of the magnetic origin by applying the magnetic field of opposite polarities as well as by measuring the MOKE above TN. They see that $\Delta\theta$ flips phase upon reversing the polarity of the magnetic field and vanishes above TN. The first term of formula (1) clearly show that this is not enough to state that the transient signal is magnetic! A simple vanishing of the equilibrium magnetization M would lead also to the disappearance of the oscillations as well as the field-induced flip of the equilibrium magnetization would π -shift phase of the MOKE response. It is not clear why the authors do not rule out this simple possibility before discussing any complex interplay of lattice and magnetism.

This controversy is typically occurring in experiments on ultrafast excitation of the magneto-acoustic dynamics (see for example Phys. Rev. Lett. 112, 147403 (2014), Phys. Rev. B 92, 020404(R) (2015), Phys. Rev. B 95, 060409(R) (2017)). The most direct way to resolve this issue is to systematically measure the magnetic field dependence and to see whether it can impact the amplitude of the oscillations in the MOKE channel. One can, for example, expect that the sufficiently strong magnetic field should be able to suppress oscillations of the magnetization. This should be no problem for the case of MnBi₂Te₄ having low TN and thus implying that the strength of the exchange interaction is only moderately large. Measuring even in the field range from 0 to 3.8 T should be sufficient to see a noticeable the impact of the external field.

Another complementary way to rule out the phonon contribution to the MOKE signal would be to study the MOKE response as a detailed function of temperature and to demonstrate that the dynamics show some signs of the critical behavior in the vicinity of TN.

Without the detailed field and temperature dependence, the scientific conclusions are rather speculative and not well justified. Thus, I suggest the authors carry out these measurements before any publication.

2. How sure are the authors that modulation of the magnetization (longitudinal dynamics) but not a simple change in its orientation (transversal dynamics)? This would also lead to a periodic modulation of the MOKE signal.

3. The authors state that modulation of the magnetization is due to direct and superexchange pathways. Can it be simply due to the modulation of magneto-crystalline anisotropy?

4. The manuscript completely overlooks progress in the ultrafast magneto-photonics studies, which has recently shown that laser-excited infrared-active phonons (~20 THz) can drive dramatic changes in magnets up to extent of picosecond switching of magnetic state and full magnetization reversal.

5. I think the authors should clearly state in the text, not in the methods, that all the measurements were performed in the canted antiferromagnetic phase with the netmagnetization (I can only assume that this is a spin-flop phase) of the MnBi₂Te₄. What is the signal in the collinear AFM phase? Could the author show the hysteresis curve in the polar MOKE geometry? Therefore Fig. 1a in the present form is misleading.

Minor comments:

1. Page 47. What is the multimodal pump-probe experiment? It looks that the authors exploit a not commonly recognized jargon.
2. Page 248. Fix reference 17.

To conclude, I find that although establishing a coupling between A_{1g} mode and magnetism is plausible, attributing the MOKE oscillations to the sub-picosecond coherent modulation of the magnetization and thus demonstration of the sub-picosecond magnetophononic control of the magnetization is not well justified and requires further validation.

Reviewer #2 (Remarks to the Author):

This manuscript by H. Padmanabhan et al. reports the observation of structural and magnetic dynamics in the layered topological insulator MnBi₂Te₄. In this work, the authors impulsively excite coherent Ag phonons with 800m pulses, which are coupled to antiferromagnetism even at equilibrium, and estimate possible changes to the exchange interactions induced by the coherent phonon displacement. The main claim of the paper is that the authors induce a coherent modulation of the magnetization through structural degrees of freedom, thus demonstrating magnetophononic manipulation in a topological material. This research area is particularly active and multiple theoretical/experimental groups are focusing on the structural control of magnetic materials.

I am two-minded about this work. In principle, the topic is interesting, and the experiments are well executed and analyzed, however I have issues with the interpretation.

The first issue I have concerns the claim of magnetophononic control. There are many things happening in a solid when illuminated with 1.55 eV pulses and purely manipulating the magnetic interaction is not the most intuitive. Unlike other experiments which take care of this issue by using THz pulses (e.g. arXiv:2101.01189), I am afraid that the leading effects on the magnetization are purely thermal. The present of hot electrons launches Ag-symmetry coherent phonons and is enough to induce a melting of the antiferromagnetic phase. The intrinsic coupling between the phonon and magnetization is just a higher order correction to these massive thermal dynamics. The authors themselves quantify the effect as "the coherent A_{1g} phonons induce a magnetization modulation of $\sim 2.5 \times 10^{-3}$ μ B under an estimated coherent A_{1g} phonon distortion of $\sim 0.4 \times 10^{-3}$ pm along the Mn-Te bond", which is a tiny effect compared to the overall magnetization.

On a slightly separate note, but still following from the previous point, I fail to see a clear-cut fingerprint of magnetophononics driving. How can the authors discriminate between a scenario in which the phonon is really driving changes in the magnetization and one in which the phonon ringing (and the coherent modulation of J) is just a byproduct of the ultrafast demagnetization due to the laser irradiation? If you launch a coherent phonon, you will modify the exchange due to the hopping terms, but you might also have other effects such as screening and thermal fluctuations which will end up dominating the dynamics.

Besides these deeper questions, I have some issues with the data presentation.

In Fig. 2 the authors overlay the magnetic reflection intensity with the frequency change of Raman mode frequency. It is clear that there is a kink in the slope, but it likely comes from magnetoelastic coupling and a change of the vibrational Grüneisen parameter. The figure implies that the phonon frequency can be read as an order parameter, and I have issues with such statement.

In Fig. 3 the authors are comparing UED, reflectivity and MOKE data. Based on the timescales of reflectivity and structural changes the authors interpret the MOKE change in terms of a two-timescale model. I regard this inappropriate because (i) the UED and optical data are taken at different fluences, and (ii) the MOKE timescale is pretty much an order parameter behavior which could be modeled in terms of a single timescale due to the temperature increase.

Finally, concerning the DFT calculations, I do not understand why the authors quote numbers for a 1pm displacement, while their estimated Ag displacements are 3 orders of magnitude lower. This would yield of course lower effective J changes. Also, what is the Lindemann limit for this material? 10% of the bond length sounds very close.

Response to Reviewer Comments

NCOMMS-21-13534-T: Sub-picosecond coherent magnetophononic coupling in MnBi₂Te₄

Contents

i.	Response summary	...	2
ii.	Summary of new time-resolved MOKE measurements	...	4
iii.	Point-by-point response – Reviewer #1	...	5
iv.	Point-by-point response – Reviewer #2	...	9
v.	Appendix – Time-resolved MOKE measurements	...	12

**Response summary**

We thank the reviewers for their careful reading of our manuscript and for their insightful comments. The
experiments and analysis motivated by the reviewers have improved the manuscript in terms of the
scientific content and clarity.

We considered all of the reviewers' comments in detail, and have carried out several new experiments,
theoretical calculations, and modeling, all of which have led to new insights on magnetophononic coupling
in MnBi_2Te_4 . In particular –

- i. We compressed the probe pulses in our time domain experiments, improving our time resolution
from ~ 300 fs to ~ 50 fs. This enabled access to *two* distinct A_{1g} phonons, as opposed to one,
originally.
- ii. Upon the recommendation of the reviewers, we carried out field-dependent transient reflectivity
($\Delta R/R$) and MOKE measurements, with applied fields from 0 to 6.8 T, across the spin-flop
transition (3.7 T) in MnBi_2Te_4 . This helped us identify the field-dependence of MOKE oscillations,
and field-dependent anomalies in coherent phonon amplitudes in $\Delta R/R$.
- iii. With the objective of separating magnetic and nonmagnetic dynamics in our MOKE measurements,
we measured transient magneto-optic polarization rotation ($\Delta\theta$) as well as ellipticity ($\Delta\eta$).
- iv. Employing a new home-built Raman spectrometer, we carried out magneto-Raman measurements
with applied fields from 0 to 9 T, across the spin-flop (3.7 T) and ferromagnetic (7.7 T) transitions
in MnBi_2Te_4 . These experiments have led to new discoveries that helped establish the main finding
in our work – coupling of phonons to the interlayer exchange coupling J^\perp .
- v. The above experimental results motivated DFT calculations of phonon modulation of the interlayer
exchange coupling J^\perp for all six Raman modes, as opposed to one A_{1g} mode in the original
submission.
- vi. We explain the new results in our work using a phenomenological model of magnetophononic
wave-mixing, with the microscopic details fleshed out via DFT calculations mentioned above.

Our field-dependent transient MOKE ($\Delta\theta$ and $\Delta\eta$) measurements show (i) qualitatively similar $\Delta\theta$ and $\Delta\eta$
dynamics, and (ii) oscillation amplitudes that track the equilibrium magnetization. The low signal levels
ultimately make the interpretation of magnetophononic coupling in MOKE inconclusive as a standalone
experiment and is thus no longer highlighted in our paper. However, it is still possible that there is a small
magnetic contribution to the observed oscillatory signal, and we note that future studies involving
resonantly pumping phonons with mid-IR and THz light may help elucidate this hypothesis.

For the scope of this paper, we find a rich spectrum of new evidence for magnetophononic coupling in our
field-dependent magneto-Raman and $\Delta R/R$ measurements –

- i. In addition to the spin-induced phonon frequency renormalization reported originally, we find
mode-specific anomalies in A_{1g} spectral weight correlated with the antiferromagnetic order
parameter.
- ii. We rationalize the above observations in terms of magnetophononic wave-mixing that activates
zone-boundary phonon modes that are otherwise forbidden by conservation of momentum.
- iii. We formulate a microscopic model based on DFT simulations in terms of phonon modulation of
the interlayer super-superexchange coupling. The model predicts a dichotomy between (out-of-
plane) A_{1g} and (in-plane) E_g modes, in remarkable agreement with our experimental results.
- iv. Our field-dependent $\Delta R/R$ traces show signatures of such magnetophononic coupling in the time
domain at sub-picosecond timescales.

Our results do show signatures of magnetophononic coupling at the sub-picosecond timescale; however,
given the more general nature of the magnetophononic coupling described in the revised manuscript, we
have dropped the word ‘sub-picosecond’ from the title.

We believe the results highlighted here are applicable to a wide range of layered and quasi-two-dimensional
magnets and open up new opportunities in the coherent manipulation of their magnetic phases. In this
context, we hope the reviewers find our new results to be of broad interest to the community.

**Summary of new time-resolved MOKE measurements**

As outlined in the Appendix of this response (see page 15, lines 5-15), we performed time-resolved
 magneto-optic Kerr rotation ($\Delta\theta$) and ellipticity ($\Delta\eta$) measurements under identical conditions, in order to
 separate magnetic dynamics from other artifacts. This is an approach that is commonly recommended in
 the literature^{1,2}. We used phonon oscillations in $\Delta R/R$ measurements to carefully calibrate $t = 0$ in all of
 our measurements, to help account for any potential drift over time. The results, (see Appendix, page 14,
 line 13 onwards) show oscillations at the frequencies of $A_{1g}^{(1)}$ and $A_{1g}^{(2)}$ modes, i. e. at 1.5 THz and 3.4 THz,
 respectively. Importantly, the oscillations are present in both $\Delta\theta$ and $\Delta\eta$, consistent with an origin in true
 magnetic dynamics, as previously argued in various references.

Next, as recommended by the reviewers, we carried out field-dependent measurements of the transient Kerr
 rotation $\Delta\theta$ (see page 15, lines 16-35). To ensure the repeatability of experiments, we used a comprehensive
 calibration procedure that accounted for small changes in $t = 0$, the overall MOKE demagnetization signal,
 and day-to-day drift. While the low signal levels make it difficult to track the amplitude of the 3.4 THz
 component ($A_{1g}^{(2)}$ phonon), the amplitude of the 1.5 THz component ($A_{1g}^{(1)}$ phonon) is extracted reliably. We
 find that the oscillation amplitude increases with the equilibrium magnetization, within the experimental
 uncertainty quantified by the error bars, as shown in Fig. A1 below (also see Fig. A3).

 **Fig. A1 Summary of $\Delta\theta$ measurements.** The amplitudes of the oscillations in $\Delta\theta$ at 1.5 THz and 3.4 THz are plotted
 as a function of applied magnetic field. The equilibrium magnetization measured using SQUID magnetometry from
 reference is overlayed on this. Error bars are the standard deviations in the overall $\Delta\theta$ amplitude summed with the
 standard deviation in fit values.

Furthermore, as outlined in the Appendix to this response (see page 15, lines 28-35), the pump-induced
 coherent phonon amplitude may itself change as a function of magnetic field (also see Fig. 2 and Fig. 4 in
 the main text). Given this, in combination with the small signal levels of the oscillations in $\Delta\theta$, we are
 unfortunately unable to separate them from more trivial sources of oscillations and conclusively attribute
 them to magnetic dynamics. In our revised manuscript, we instead focus on the various other manifestations
 of magneto-phononic coupling in MnBi_2Te_4 (see page 2, lines 34-43).

**Point-by-point response – Reviewer #1**

The manuscript *H. Pardmanabhan et al.* reports on demonstrating a coherent coupling between displacively
excited optical phonon mode and magnetization in the topological insulator MnBi₂Te₄ as revealed by time-
resolved two-color pump-probe experiment. These conclusions are based on observing a modulation of the
magneto-optical Kerr signal by coherent oscillations corresponding to the Ag Raman-active phonon mode.
This is further corroborated by static Raman spectroscopy data showing a correlation between the onset of
the magnetic order and shift in the frequency of the Ag mode as well as by DFT calculations.

Although I find the Raman spectroscopy data convincing and theory somewhat interesting, the
interpretation of the time-resolved experiment, which is the main subject of the study, raises serious
concerns. Below I provide a detailed list of major and minor comments to the current version of the
manuscript.

We thank the reviewer for their critical reading of our manuscript. The concerns raised here motivated us
to carry out a comprehensive set of field-dependent pump-probe and steady-state spectroscopy
measurements.

In our revised manuscript, we present convincing experimental evidence for coherent coupling between
optical phonons and interlayer exchange. We rationalize our observations using a simple model of
magnetophononic wave-mixing and show a remarkable agreement between our experimental results and
DFT calculations. Our field-dependent MOKE measurements are unfortunately inconclusive, as explained
earlier (page 4) and in the Appendix at the end (pages 12-17) and are no longer highlighted in the
manuscript. Despite the major revisions to our manuscript, we hope the reviewer appreciates our meticulous
follow-up to their queries and suggestions, the rich spectrum of new evidence for magnetophononic
coupling, and broad applicability thereof in the growing field of quasi-two-dimensional magnetic materials.

**Major comments:**

1. The authors rely on fact that the transient MOKE signal $\Delta\theta$ is proportional to the magnetization ΔM ,
via the polar Kerr effect. However, it is commonly known that one of the proportionality factors relies
on the intensity of the incident/reflected light R, such that:

$$\Delta\theta \sim \Delta R * M + R * \Delta M \quad (1)$$

implying that modulation of the light intensity ΔR can contribute to the transient Kerr
rotation. That is the reason why the polarization rotation can be in principle be also subjected to a
modulation due to phonon modes. It is known that even in the absence of magnetization the time-
resolved polarization rotation can be modulated by the coherent acoustic and optical phonons (e.g. *J.*
*Appl. Phys.* **114**, 093513 (2013)) due to the transient birefringence.

The authors make a fair attempt to clarify whether the oscillations of the magnetic origin by applying
the magnetic field of opposite polarities as well as by measuring the MOKE above TN. They see that
$\Delta\theta$ flips phase upon reversing the polarity of the magnetic field and vanishes above TN. The first term
of formula (1) clearly show that this is not enough to state that the transient signal is magnetic! A simple
vanishing of the *equilibrium magnetization* M would lead also to the disappearance of the oscillations
as well as the field-induced flip of the *equilibrium magnetization* would π -shift phase of the MOKE
response. It is not clear why the authors do not rule out this simple possibility before discussing any
complex interplay of lattice and magnetism.

We thank the reviewer for this comment. In the Appendix (page 12, line 11 onwards), we outline all
the possible sources of nonmagnetic contributions to $\Delta\theta$, and describe our strategies to account for
them. In particular, as the reviewer points out, we do eliminate contributions due to transient

birefringence. Following our original submission, we have carried out several new measurements to
account for nonmagnetic artifacts, including the $M\Delta R$ -type term highlighted by the reviewer above.
These are outlined below.

This controversy is typically occurring in experiments on ultrafast excitation of the magneto-acoustic
dynamics (see for example *Phys. Rev. Lett.* **112**, 147403 (2014), *Phys. Rev. B* **92**, 020404(R) (2015),
*Phys. Rev. B* **95**, 060409(R) (2017)). The most direct way to resolve this issue is to systematically
measure the magnetic field dependence and to see whether it can impact the amplitude of the
oscillations in the MOKE channel. One can, for example, expect that the sufficiently strong magnetic
field should be able to suppress oscillations of the magnetization. This should be no problem for the
case of MnBi₂Te₄ having low TN and thus implying that the strength of the exchange interaction is
only moderately large. Measuring even in the field range from 0 to 3.8 T should be sufficient to see a
noticeable the impact of the external field. Another complementary way to rule out the phonon
contribution to the MOKE signal would be to study the MOKE response as a detailed function of
temperature and to demonstrate that the dynamics show some signs of the critical behavior in the
vicinity of TN.

We acknowledge the importance of eliminating nonmagnetic contributions to the $\Delta\theta$ signals and the
controversy in the literature. With the objective of separating true magnetic dynamics from
nonmagnetic artifacts, we employ a commonly used strategy in the literature^{1,3,4}, namely the
measurement of the magneto-optic Kerr rotation $\Delta\theta$ as well as ellipticity $\Delta\eta$ (see page 15, lines 5-15
and equation 8). While the two exhibit qualitatively similar dynamics (see Fig. A2), consistent with
magnetic dynamics, this is insufficient proof for magnetic dynamics, as argued in the Appendix (see
page 15, lines 5-15).

Next, as recommended by the reviewer, we carried out field-dependent $\Delta\theta$ measurements, the details
of which can be found in the Appendix (page 15, line 16 onwards). Our results are summarized in Fig.
A3. The oscillation amplitude tracks the equilibrium magnetization, consistent with an $M\Delta R$ -type
contribution to $\Delta\theta$. As we explain in the Appendix (page 15, lines 29-35), the Raman and normalized
coherent phonon amplitudes also exhibit a significant field-dependence, which precludes a direct
comparison to the equilibrium M . Ultimately, given the low signal levels and significant experimental
uncertainty, there is insufficient evidence to conclude that phonons coherently modulate the
magnetization.

Without the detailed field and temperature dependence, the scientific conclusions are rather speculative
and not well justified. Thus, I suggest the authors carry out these measurements before any publication.

We thank the reviewer for bringing this important detail to our notice and helping us improve the
integrity of our work. Based on our improved understanding of the MOKE results, we have removed
this from our revised manuscript. We instead focus on a variety of new, convincing signatures of
magnetophononic coupling found via field-dependent Raman and pump-probe spectroscopy, as
summarized earlier (page 2, lines 34-43).

2. How sure are the authors that modulation of the magnetization (longitudinal dynamics) but not a simple
change in its orientation (transversal dynamics)? This would also lead to a periodic modulation of the
MOKE signal.

Please refer to the above response (lines 18-33).

3. The authors state that modulation of the magnetization is due to direct and superexchange pathways.
Can it be simply due to the modulation of magneto-crystalline anisotropy?

Please refer to the above response (page 6, lines 18-33). On the other hand, our new results, highlighted
in Fig. 1 and 2 of the main text show a clear correlation between the phonon spectral weight and
antiferromagnetic order parameter, rather than the overall orientation of spins, as one might expect with
the magneto-crystalline anisotropy. This is now explicitly shown in the schematics of the various
magnetic phases shown in Fig. 2. In particular, the primary difference between the ground state AFM
and high-field FM states is the interlayer magnetic coupling, with the spin orientations being along the
c axis in both cases, and the intralayer coupling remaining unchanged. That the observed results are a
direct consequence of phonons coupling to the interlayer exchange is also confirmed by our DFT
calculations, which show that only A_{1g} modes strongly modulate J^\perp , as shown in Fig. 3, and in the
main text in page 8, line 17 onwards.

- 4. The manuscript completely overlooks progress in the ultrafast magneto-photonics studies, which has
recently shown that laser-excited infrared-active phonons (~20 THz) can drive dramatic changes in
magnets up to extent of picosecond switching of magnetic state and full magnetization reversal.

These studies are now explicitly cited in the main text in page 14, lines 8-10. The results highlighted in
our work motivate phonon pumping approaches such as resonant THz excitation to drive magnetic and
topological phase transitions in MnBi_2Te_4 . In the discussion section in the main text (see page 14, lines
10-16), we highlight recent theoretical work that explores these possibilities.

- 5. I think the authors should clearly state in the text, not in the methods, that all the measurements were
performed in the canted antiferromagnetic phase with the net magnetization (I can only assume that
this is a spin-flop phase) of the MnBi_2Te_4 . What is the signal in the collinear AFM phase? Could the
author show the hysteresis curve in the polar MOKE geometry? Therefore Fig. 1a in the present form
is misleading.

The reviewer is correct that the measurements in our original submission were carried out in a spin-
flop phase. There were also no A_{1g} oscillations observed in $\Delta\theta$ at 0 T, consistent with an $M\Delta R$ type
signal, as mentioned earlier (page 6, lines 18-33).

In the revised manuscript, we include the field-dependent signal for both the Raman spectroscopy (0 to
9 T, see Fig. 1 and Fig. 2 in the main text) as well as $\Delta R/R$ pump-probe measurements (0 to 6.8 T, see
Fig. 4 in the main text). There is no observed hysteresis effect, i. e. the signal at 0 T is repeatable across
field sweeps, consistent with the antiferromagnetic ground state of MnBi_2Te_4 . In order to explicitly
convey the magnetic states, we have included schematics of the spin structures in all the probed
magnetic states in Fig. 2 of the main text.

Minor comments:

- 1. Page 47. What is the multimodal pump-probe experiment? It looks that the authors exploit a not
commonly recognized jargon.

By ‘multimodal’, we refer to the fact that the pump-probe experiments were carried out with similar
pump excitation (namely, 800 nm, 1.55 eV), but multiple, complementary probes ($\Delta R/R$, $\Delta\theta$, $\Delta\eta$,
electron diffraction) that access different degrees of freedom. Our intent was to use nomenclature
analogous to that used in optical microscopy in the case of multiple simultaneous probing modalities.
The word ‘multimodal’ has now been removed from the manuscript.

- 2. Page 248. Fix reference 17.

We have fixed this, as well as some other typos in the references.

To conclude, I find that although establishing a coupling between A1g mode and magnetism is plausible,
attributing the MOKE oscillations to the sub-picosecond coherent modulation of the magnetization and thus
demonstration of the sub-picosecond magnetophononic control of the magnetization is not well justified
and requires further validation.

Upon carrying out the experiments recommended by the reviewer, as well as additional checks such as $\Delta\theta$
vs. $\Delta\eta$ and field-dependent Raman spectroscopy, we are unable to conclusively magnetophononic coupling
manifesting as magnetization oscillations in $\Delta\theta$. This has hence been removed from the paper. On the other
hand, we do find convincing evidence for A_{1g} phonons coupling to the interlayer exchange J^\perp from several
other measurements and theoretical calculations, as highlighted earlier (page 2, lines 34-43) and in the
manuscript. We hope the substantial revisions made to the manuscript provide sufficient evidence for our
claim of magnetophononic coupling.

**Point-by-point response – Reviewer #2**

This manuscript by H. Padmanabhan et al. reports the observation of structural and magnetic dynamics in
the layered topological insulator MnBi₂Te₄. In this work, the authors impulsively excite coherent Ag
phonons with 800m pulses, which are coupled to antiferromagnetism even at equilibrium, and estimate
possible changes to the exchange interactions induced by the coherent phonon displacement. The main
claim of the paper is that the authors induce a coherent modulation of the magnetization through structural
degrees of freedom, thus demonstrating magnetophononic manipulation in a topological material. This
research area is particularly active and multiple theoretical/experimental groups are focusing on the
structural control of magnetic materials.

I am two-minded about this work. In principle, the topic is interesting, and the experiments are well
executed and analyzed, however I have issues with the interpretation.

We appreciate the reviewer’s interest in the topic of our manuscript and thank them for their comments on
the execution of experiments and analysis. As mentioned earlier (page 4, lines 22-27), our experiments on
coherent modulation of magnetism were not conclusive. On the other hand, through our field-dependent
Raman spectroscopy and pump-probe measurements, we have found new, convincing evidence for
coupling of A_{1g} phonons with antiferromagnetism, as outlined in the summary (page 2, lines 34-43). This
is indeed found to be the consequence of A_{1g} phonons modulating the exchange coupling as in the original
manuscript. This experimentally manifests as the optical generation of zone-boundary phonons, which are
otherwise forbidden by momentum conservation (see page 4, line 14 onwards in the main text). We hope
our revised manuscript and the extensive new experiments and evidence of coupling of A_{1g} modes to the
interlayer exchange interaction are of interest to the reviewer.

The first issue I have concerns the claim of magnetophononic control. There are many things happening in
a solid when illuminated with 1.55 eV pulses and purely manipulating the magnetic interaction is not the
most intuitive. Unlike other experiments which take care of this issue by using THz pulses (e.g.
arXiv:2101.01189), I am afraid that the leading effects on the magnetization are purely thermal. The present
of hot electrons launches Ag-symmetry coherent phonons and is enough to induce a melting of the
antiferromagnetic phase. The intrinsic coupling between the phonon and magnetization is just a higher order
correction to these massive thermal dynamics. The authors themselves quantify the effect as “the coherent
A_{1g} phonons induce a magnetization modulation of $\sim 2.5 \times 10^{-3} \mu\text{B}$ under an estimated coherent A_{1g}
phonon distortion of $\sim 0.4 \times 10^{-3} \text{ pm}$ along the Mn-Te bond”, which is a tiny effect compared to the overall
magnetization.

We thank the reviewer for their critical comments. We agree with the reviewer – the dominant effect upon
above-bandgap excitation is indeed the generation of carriers, and subsequent thermalization. However, we
can clearly separate the signal into the exponential background signal, nominally related to the thermal
dynamics, and the oscillatory signal. Although the background signal is larger, this does not preclude our
ability to analyze the oscillatory signal.

The carrier-lattice thermalization does result in some degree of ultrafast demagnetization, however, our
fluence-dependent measurements in the Appendix Fig. A4 (page 17, lines 7-16) show that the chosen pump
fluence of $100 \mu\text{J}/\text{cm}^2$ is well below the threshold to fully melt the magnetization, which is over $300 \mu\text{J}/\text{cm}^2$.

Furthermore, even though it is true that the displacively excited coherent phonons are small compared to
the overall carrier and thermal dynamics, in our analysis in the revised manuscript, we focus on signatures

of magnetophononic coupling within this small signal. Such methods have been used to provide insight into
low-energy degrees of freedom in previous publications⁵⁻⁸.

On a slightly separate note, but still following from the previous point, I fail to see a clear-cut fingerprint
of magnetophononics driving. How can the authors discriminate between a scenario in which the phonon
is really driving changes in the magnetization and one in which the phonon ringing (and the coherent
modulation of J) is just a byproduct of the ultrafast demagnetization due to the laser irradiation? If you
launch a coherent phonon, you will modify the exchange due to the hopping terms, but you might also have
other effects such as screening and thermal fluctuations which will end up dominating the dynamics.

Please see previous response (page 9, lines 13-14, page 4, and Appendix).

In the revised manuscript, we find time domain signatures of magnetophononic coupling using magnetic
field-dependent $\Delta R/R$ measurements. Here, we quantify the effect of carrier excitation by measuring the
fluence dependence in SI Section S5 and Fig. S5. Assuming the pump-induced carrier density is
proportional to the excitation fluence, we show that maximum amplitude of the $\Delta R/R$ signal scales linearly
with fluence, as do the coherent phonon amplitudes. Based on this, we normalize the $\Delta R/R$ traces in Fig. 4
in the main text to their maximum amplitudes. We then plot field-induced changes in coherent phonon
amplitudes in the normalized traces, as outlined in page 10, line 21 onwards, in the main text.

Besides these deeper questions, I have some issues with the data presentation.

In Fig. 2 the authors overlay the magnetic reflection intensity with the frequency change of Raman mode
frequency. It is clear that there is a kink in the slope, but it likely comes from magnetoelastic coupling and
a change of the vibrational Grüneisen parameter. The figure implies that the phonon frequency can be read
as an order parameter, and I have issues with such statement.

We agree with the reviewer – the phonon frequency cannot be used as a magnetic order parameter. Our
chosen method of data presentation was motivated by the model of spin-induced phonon frequency
renormalization explained in SI Section S4 and Equation 4, according to which the phonon frequency
renormalization is proportional to the magnetic order parameter, i. e. $\Delta\nu/\nu \propto \langle S^2 \rangle$. We convey this by
plotting the phonon frequency change and magnetic Bragg peak intensity on a common x axis but separate
y axes. In Fig. 2 of our revised manuscript, we employ a similar method, highlighting the correlation
between the phonon integrated intensity and the antiferromagnetic order parameter by plotting them on a
common x axis but separate y axes. We show in Fig. 3 and in page 4, line 14 onwards in the main text that
the anomalous spectral weight is in fact directly related to the antiferromagnetic order parameter through a
wave-mixing process that activates zone-boundary modes. Here, again, we do not claim that the phonon
spectral weight can be read as the magnetic order parameter – this is explicitly conveyed by using separate
y axes with distinct labels.

The observed effects may indeed be classified under the umbrella of magnetoelastic coupling, however, as
highlighted in our manuscript (page 8, line 17 onwards), we are able to experimentally resolve a specific
case of interest, namely the modulation of the interlayer exchange J^\perp , selectively by A_{1g} phonons. This
mode-selectivity is further supported by the lack of such a signature in other phonons (see SI Section S3)
as well as our DFT calculations in Fig. 3.

In Fig. 3 the authors are comparing UED, reflectivity and MOKE data. Based on the timescales of
reflectivity and structural changes the authors interpret the MOKE change in terms of a two-timescale
model. I regard this inappropriate because (i) the UED and optical data are taken at different fluences, and

(ii) the MOKE timescale is pretty much an order parameter behavior which could be modeled in terms of a
single timescale due to the temperature increase.

We agree with the reviewer – the demagnetization dynamics need to be modeled by considering the
transient lattice temperature and the magnetic order parameter’s dependence on it. In our revised
manuscript, we no longer compare the UED thermalization timescale with the timescale of
demagnetization, and instead simply use it to establish a timescale of phonon thermalization (see Fig. 4f
and page 13, line 1 onwards in the main text).

The large fluence used in the MeV-UED measurements is a consequence of the low signal levels. A fluence
of the order of 5 mJ/cm^2 was required to obtain usable data. In the revised manuscript, we show the Debye-
Waller time constant as a function of fluence (see SI Section S6 and Fig. S6d) and note that there is only a
small change within the measured range of fluences. Based on the expectation of increased phonon-phonon
scattering at higher fluences, our UED measurements set a lower bound for the phonon thermalization
timescale (see page 8, lines 18-29 in the SI). Since our objective in the revised manuscript is only to
establish a timescale of phonon thermalization, rather than argue for a two-timescale model, we consider
this an appropriate experimental yardstick.

Finally, concerning the DFT calculations, I do not understand why the authors quote numbers for a 1pm
displacement, while their estimated Ag displacements are 3 orders of magnitude lower. This would yield
of course lower effective J changes. Also, what is the Lindemann limit for this material? 10% of the bond
length sounds very close.

The goal of our DFT calculations was to characterize how the phonons of interest change the exchange
couplings. The low energy scales associated with the exchange couplings ($<1 \text{ meV}$) and the practical limits
of DFT energy convergence necessitated phonon displacements of the order of 1 pm to see discernible
changes in the calculated exchange couplings. In the revised manuscript, we only make use of the slope
dJ^\perp/du , rather than the absolute change in J^\perp . We also note that a displacement of 1 pm has been shown
to be accessible via resonant excitation using THz sources⁹. A displacement of 1 pm corresponds to a change
of $10^{-12} \text{ m}/4 \times 10^{-10} \text{ m} \sim 0.25\%$ in the bond length and is almost two orders of magnitude smaller than the
typical Lindemann criterion of 10-20% of the lattice constant. The Lindemann criterion for MnBi_2Te_4 is
not known, but Bi was experimentally found¹⁰ to have a Lindemann criterion $>9\%$.

Appendix – Time-resolved MOKE measurements

The polar magneto-optic Kerr effect (MOKE) is generally used to measure the polarization rotation or
 ellipticity upon reflection off a sample where the magnitude of rotation and ellipticity is related to the net
 magnetization of the sample. In this sense, MOKE has generally been used to study the properties of ferro-
 and ferrimagnets, but the technique has also seen use in understanding the magneto-optical properties of
 antiferromagnetic materials such as MBT¹¹. Additionally, time-resolved MOKE is useful in understanding
 the magnetization dynamics on time-scales fundamental to the exchange interactions within the material¹².
 In light of the evidence of magnetophononic coupling described in the main text, we use time-resolved
 MOKE to look for signatures of similar coupling in the time-domain magnetization dynamics of MnBi₂Te₄
 across various phases of magnetic ordering.

Methods to eliminate or characterize non-magnetic components of MOKE signal

We carry out a series of checks to ensure that the coherent oscillations in the transient Kerr signal $\Delta\theta =$
 $(\Delta\theta_H - \Delta\theta_{-H})/2$ are magnetic in origin. In general, we are interested in the complex MOKE angle (Voigt
 vector) defined as

$$15 \quad \tilde{\Theta}(t) = \theta(t) + i\eta(t), \quad (1)$$

where the real component $\theta(t)$ and the imaginary component $\eta(t)$ are the time-dependent the Kerr angle
 and ellipticity respectively – these two components are the observable quantities that are measured by
 conventional Kerr rotation and Kerr ellipticity measurements¹. Phenomenologically, the Voigt vector can
 also be related to material parameters via

$$20 \quad \tilde{\Theta}(t) = \tilde{N}(t) + \tilde{F}(t) \cdot \vec{M}(t), \quad (2)$$

Where $\tilde{N}(t)$ is the non-magnetic contribution to the MOKE signal, $\tilde{F}(t)$ is the effective Fresnel coefficient
 corresponding to diagonal terms in the dielectric tensor³, and $\vec{M}(t)$ is the sample magnetization. For clarity,
 we will continue the following discussion on isolating true magnetic signals by focusing on the real part of
 the MOKE signal, $\theta(t)$, noting that similar results can be derived for the imaginary part. That is,

$$25 \quad \theta(t) = N(t) + F_1(t) \cdot \vec{M}(t), \quad (3)$$

where $\tilde{F}(t) = F_1(t) + iF_2(t)$. In pump-probe spectroscopy, we are interested in the transient pump-
 induced changes in the MOKE signal $\Delta\theta(t)$, giving us the expression

$$28 \quad \theta(t) = \theta_0 + \Delta\theta(t) = N_0 + \Delta N(t) + F_{1,0} \cdot \vec{M}_0 + F_{1,0} \cdot \Delta\vec{M}(t) + \Delta F_1(t) \cdot \vec{M}_0. \quad (4)$$

The terms $\theta_0, N_0, F_{1,0}$, and \vec{M}_0 are the steady-state values for their respective quantities, and $\Delta N(t), \Delta\vec{M}(t)$,
 and $\Delta F_1(t)$ are the transient pump-induced changes for the respective quantities. It is immediately apparent
 that the MOKE signal generally contains contributions to the signal that do not correspond to true
 magnetization dynamics, $\Delta\vec{M}(t)$. We categorize these parasitic contributions into three general sources and
 outline the methods we used to eliminate or minimize them.

I. Nonmagnetic terms ($\Delta N(t) = N_0 + \Delta N(t)$)

The major source (I) is the non-magnetic contribution, $N(t) = N_0 + \Delta N(t)$, which is symmetric with
 respect to magnetization reversal. This can originate from, for example, pump-induced birefringence,
 dichroism, or two-photon absorption and generally corresponds to the time-dependent off-diagonal terms

1 in the dielectric tensor¹. These non-magnetic contributions can be corrected by defining the true MOKE
 2 signal as $\theta_{MOKE}(t) = \frac{1}{2}[\theta(+M(t)) - \theta(-M(t))]$. This leaves us with the expression

$$3 \quad \theta_{MOKE}(t) = F_{1,0} \cdot \vec{M}_0 + F_{1,0} \cdot \Delta\vec{M}(t) + \Delta F_1(t) \cdot \vec{M}_0. \quad (5)$$

In MBT, we achieved this by measuring the Kerr signal at equal and opposite magnetic fields, with the
 applied field along the c-axis.

**II. Kerr rotation detection ($\Delta R(t) \cdot \theta_0$)**

The second source (II) of parasitic signal comes from the detection scheme utilized for polar MOKE
 described in the main text and reference¹³. Namely, the raw MOKE signal which is measured directly is the
 difference between the intensity of reflected s-polarized and p-polarized light detected by two photodiodes.
 The Jones matrix formalism is used to show that the measured signal corresponds to the following:

$$11 \quad I_{MOKE} = I_p - I_s = 2R \cdot \theta,$$

$$12 \quad I_{MOKE}(t) = I_0 + \Delta I_{MOKE}(t) = 2[R_0 \cdot \theta_0 + \Delta R(t) \cdot \theta_0 + R_0 \cdot \Delta\theta(t)] \quad (6)$$

13 where R_0 is the equilibrium reflectivity and θ_0 is the relative equilibrium polarization rotation angle with
 14 respect to the polarization analyzer (Wollaston prism)¹³. In this case, the parasitic signal corresponds to the
 15 $\Delta R(t) \cdot \theta_0$ term. This signal component can be eliminated by balancing the probe intensities onto the
 16 photodiodes in the absence of the pump modulation. This ensures that

$$17 \quad I_{MOKE}(t < 0) = I_0 = 2R_0 \cdot \theta_0 = 0,$$

$$18 \quad \Rightarrow \theta_0 = 0 \text{ and } I_{MOKE}(t) = \Delta I_{MOKE}(t) = 2R_0 \cdot \Delta\theta(t). \quad (7)$$

**III. Time-odd nonmagnetic contributions ($\Delta F_1(t) \cdot \vec{M}_0$)**

The third source (III) originates from the last term in the equation (5), $\Delta F_1(t) \cdot \vec{M}_0$. This term gives a time-
 dependent signal that is intrinsic in standard polar MOKE experiments and relates to the diagonal terms in
 the dielectric tensor similarly measured by time-resolved reflectivity measurements $\Delta R(t)$. Furthermore,
 this term is odd with respect to magnetization reversal and is therefore not eliminated by the method above.
 It should be noted that $\Delta F_1(t)$ does not rigorously equate to $\Delta R(t)$, so the two signals do not necessarily
 match. There is no systematic method to fully eliminate this component of the MOKE signal, but we have
 applied two methods to estimate the relative magnitude of this parasitic signal: (i) we track the deviations
 in temporal profiles of the Kerr rotation dynamics $\Delta\theta(t)$ and the Kerr ellipticity dynamics $\Delta\eta(t)$ under
 equivalent conditions, and (ii) we track deviations in the magnetic field dependence of the MOKE signal
 amplitudes $\Delta\theta(t)$ from the equilibrium magnetization M_0 . Method (i) is derived from the practical relation

$$30 \quad \frac{\Delta\tilde{\theta}(t)}{\tilde{\theta}_0} = \frac{\Delta M(t)}{M_0} \Rightarrow \frac{\Delta\theta(t)}{\theta_0} = \frac{\Delta\eta(t)}{\eta_0}, \quad (8)$$

where

$$32 \quad \frac{\Delta\theta(t)}{\theta_0} = \frac{\Delta M(t)}{M_0} + \frac{\Delta F_1(t)}{F_{1,0}},$$

$$33 \quad \frac{\Delta\eta(t)}{\eta_0} = \frac{\Delta M(t)}{M_0} + \frac{\Delta F_2(t)}{F_{2,0}}.$$

This relationship holds if the contribution from the effective Fresnel coefficient to the MOKE signal is
 negligible. However, we emphasize that *this condition is necessary but insufficient* to show that the MOKE
 signal is directly related to true magnetization dynamics. This is because if a modulation of the real part of
 the dielectric function occurs on a time-scale τ , the imaginary part of the dielectric function must also be
 modulated on the same time-scale τ . However, we note that the amplitudes of these signals are generally
 not equal and this relation has served as a practical gauge for materials with large magneto-optical
 responses^{1,2}. Moreover, previous work³ has proposed of the following practical hierarchy:

$$8 \quad \frac{\Delta\theta(t)}{\theta_0} \approx \frac{\Delta\eta(t)}{\eta_0} \gg \frac{\Delta R(t)}{R_0}, \frac{\Delta T(t)}{T_0} \quad (9)$$

on the basis that the relative modulations of off-diagonal terms of the dielectric function are generally much
 larger than those of the diagonal terms. We discuss how this relationship hierarchy applies to our results in
 the following section.

MOKE results

 **Fig. A2 Time-resolved magneto-optic Kerr rotation vs. ellipticity.** **a**, Pump-induced Kerr rotation $\Delta\theta$ and ellipticity
 $\Delta\eta$ as a function of time delay, normalized to values at 1.5 ps. The black lines are biexponential decay fits to the
 experimental data. **b**, **c**, The residual $\Delta\theta$ and $\Delta\eta$, respectively, upon subtracting the black lines in panel a. The dots are
 experimental datapoints, the solid lines are fits to the sum of two decaying sinusoidal functions.

The time-domain MOKE signal $\Delta\theta_{MOKE}(t) = \frac{1}{2}[\Delta\theta(\vec{M}(t)) - \Delta\theta(-\vec{M}(t))]$ at 3.8 T is plotted in Fig.
 A2a. We observe oscillations with frequencies that correspond to the $A_{1g}^{(1)}$ and $A_{1g}^{(2)}$ phonons, as seen in the
 residual plot in Fig. A2b. We actively eliminated the sources of non-magnetic signal listed above with the

exception of source (III), as this component of the signal is intrinsic to the pump-modulated MOKE angle
 itself. Ultimately, we show that the dominant exponential dynamics observed in the MOKE signal are
 representative of true demagnetization dynamics, but we are unable to conclusively resolve whether the
 oscillations are indicative of true magnetization dynamics or originate from $\Delta F_1(t) \cdot \vec{M}_0$.

To address the parasitic source (III), $\Delta F_1(t) \cdot M_0$, we investigate the two methods discussed above. In Fig.
 A2a, we show the time-resolved Kerr ellipticity signal, $\Delta\eta(t) = \frac{1}{2} [\Delta\eta(\vec{M}(t)) - \Delta\eta(-\vec{M}(t))]$, overlaid
 with the Kerr rotation signal, with both traces normalized to their values at 1.5 ps, where the $\Delta R/R$ is
 measured to be 0. In Figs. A2b and A2c, we show the residual oscillations in $\Delta\theta$ and $\Delta\eta$, respectively,
 which look qualitatively similar. Though we were not able to directly measure the steady-state Kerr rotation
 and ellipticity to obtain $\Delta\theta/\theta$ and $\Delta\eta/\eta$, we use methods similar to reference¹⁴ to show that there is
 significant overlap in the temporal dynamics. This is suggestive that the Kerr rotation signal is
 representative of true magnetization dynamics; however, this is insufficient evidence that the oscillations
 are truly magnetic in origin as discussed above. Furthermore, we note that the oscillations are small relative
 to the dominant exponential dynamics, $\Delta\theta_{osc} \ll \Delta\theta_{total}$, so we *cannot* use the practical hierarchy of
 Equation (9) as proposed in reference³.

Next, to verify the magnetic origin of the $\Delta\theta_{osc}$, we measure it as a function of magnetic field as shown in
 Fig. A3a and compare its field-dependent amplitude to that of the equilibrium magnetic moment, M_0 . To
 do this, we fit the field-dependent traces in Fig. A3a to the following function:

$$19 \quad \Delta\theta(t) = \sum_i A_i \cdot e^{-\frac{t}{\gamma_i}} \cdot \cos(\omega_i t + \phi_i) + B(t),$$

where the sum is taken over the relevant oscillation frequencies observed in transient reflectivity ($A_{1g}^{(1)}$ and
 $A_{1g}^{(2)}$). The oscillations are fit as damped oscillators given by the first term with A_i, γ_i, ω_i , and ϕ_i all acting
 as free parameters. The second term $B(t)$ is the non-oscillatory background signal dominated by ultrafast
 demagnetization fit to a multiexponential:

$$24 \quad B(t) = \frac{B_1}{2} \left[1 - \operatorname{erf}\left(\frac{t - t_0}{\tau_1}\right) \right] \cdot \left(B_2 e^{\frac{t}{\tau_2}} + B_3 e^{\frac{t}{\tau_3}} \right) + C,$$

The factor containing the error function represents the cross-correlation of the pump and probe pulses, and
 C is a constant offset. We show oscillatory component of the data and respective fits in Fig A3b. While the
 small signal levels make it difficult to estimate the amplitude at the frequency $A_{1g}^{(2)}$, the slower oscillation
 at the frequency of $A_{1g}^{(1)}$ is reliably extracted. In Fig A3c, we compare the amplitudes of these oscillatory
 components of the MOKE signal to M_0 . Focusing on the 1.5 THz oscillation, we find that it tracks the
 equilibrium magnetization within the experimental uncertainty. We also note that the pump-induced
 coherent phonon amplitude may itself change as a function of magnetic field. Our magneto-Raman
 measurements show that the Raman susceptibility of the $A_{1g}^{(1)}$ and $A_{1g}^{(2)}$ modes decrease with increasing
 magnetic field, tracking the antiferromagnetic order parameter. The coherent phonon amplitudes in our
 $\Delta R/R$ measurements also exhibit such an effect, albeit only after normalization to the initial photocarrier
 density.

**Fig. A3 Field-dependent magneto-optic Kerr rotation.** **a**, Traces of the time-resolved pump-induced Kerr rotation
 $\Delta\theta$ at various magnetic fields. The black line is a biexponential decay fit to the data at 6.8 T. **b**, The residual $\Delta\theta$ upon
 subtracting biexponential decay fits from the data in panel a. The circles are experimental datapoints, and the solid
 grey lines are fits to the sum of two decaying sinusoidal functions. **c**, The amplitude of oscillations extracted from fits
 to the residual $\Delta\theta$ in panel b. The grey line is the magnetization measured using SQUID magnetometry from reference.
 Error bars are the standard deviations in the overall $\Delta\theta$ amplitude summed with the standard deviation in fit values.

In this context, given the low signal levels and large experimental uncertainty indicated by our error bars,
 we are unable to attribute the oscillations in $\Delta\theta$ to magnetic dynamics. They are instead nominally
 consistent with a $\Delta F_1(t) \cdot \vec{M}_0$ -type contribution to $\Delta\theta$.

In summary, we have systematically measured the time-resolved MOKE response of MBT using a two-
 color pump-probe scheme (1.55 eV and 1.2 eV pump and probe photon energies respectively) as described
 in Methods. We describe the measures used to ensure that the MOKE signals are representative of true
 magnetization dynamics. Oscillations at 1.47 THz and 3.4 THz appear in both the Kerr rotation and
 ellipticity, which we attribute to some signal modulation due to the $A_{1g}^{(1)}$ and $A_{1g}^{(2)}$ phonons observed in
 $\Delta R(t)/R$. However, we are unable to attribute the oscillations in the MOKE signal to true magnetization

dynamics. We emphasize that time-resolved MOKE experiments with small signal components should be
carefully scrutinized as conventional assumptions underlying the direct relation between MOKE and
magnetization may not hold. Ultimately, it is still possible that there is a small magnetic contribution to our
measured signal. Further studies that may validate this hypothesis involve resonantly pumping phonons in
MBT with mid-IR and THz light to coherently generate oscillations to large amplitudes and minimize
background signals originating from thermal effects.

Melting threshold

We explicitly verify that the pump fluence of $100 \mu\text{J}/\text{cm}^2$ used in our experiments does not fully melt the
magnetic order. To do this, we measure the pump-induced demagnetization by detecting the pump-induced
magneto-optic Kerr rotation, $\Delta\theta$, as a function of fluence. The measurements are carried out at 30 K, 1 T,
where the equilibrium magnetization is $\sim 0.3 \mu_B$. Our results, shown in Fig. A4 indicate that the fluence
threshold to completely melt the magnetic order at 1 T is over $300 \mu\text{J}/\text{cm}^2$.

**Fig. A4 Fluence-dependence of ultrafast demagnetization.** **a**, The pump-induced magneto-optic Kerr rotation $\Delta\theta$
at 30 K, 1 T, as a function of pump fluence. **b**, The maximum $|\Delta\theta|$ in panel a as a function of fluence. The dashed line
is a guide to the eye, denoting the saturation demagnetization.

1. Antoinette, J. & Rohit, P. *Optical techniques for solid state materials characterization*.
2. Wang, J. *et al.* Memory effects in photoinduced femtosecond magnetization rotation in ferromagnetic GaMnAs. *Appl. Phys. Lett.* **94**, 92–95 (2009).
3. Wang, J. *et al.* Ultrafast magneto-optics in ferromagnetic III-V semiconductors. *J. Phys. Condens. Matter* **18**, (2006).
4. Koopmans, B., Van Kampen, M., Kohlhepp, J. T. & De Jonge, W. J. M. Ultrafast magneto-optics in nickel: magnetism or optics? *Phys. Rev. Lett.* **85**, 844–847 (2000).
5. Melnikov, A. *et al.* Coherent Optical Phonons and Parametrically Coupled Magnons Induced by Femtosecond Laser Excitation of the Gd(0001) Surface. *Phys. Rev. Lett.* **91**, 227403 (2003).
6. Lee, M. C. *et al.* Strong spin-phonon coupling unveiled by coherent phonon oscillations in Ca₂RuO₄. *Phys. Rev. B* **99**, 1–5 (2019).
7. Gerber, S. *et al.* Femtosecond electron-phonon lock-in by photoemission and x-ray free-electron laser. *Science (80-.)*. **357**, 71–75 (2017).
8. Harter, J. W. *et al.* Evidence of an Improper Displacive Phase Transition in Cd₂Re₂O₇ via Time-Resolved Coherent Phonon Spectroscopy. *Phys. Rev. Lett.* **120**, 47601 (2018).
9. Kozina, M. *et al.* Terahertz-driven phonon upconversion in SrTiO₃. *Nat. Phys.* (2019) doi:10.1038/s41567-018-0408-1.
10. Sokolowski-Tinten, K. *et al.* Femtosecond x-ray measurement of coherent lattice vibrations near the lindemann stability limit. *Nature* **422**, 287–289 (2003).
11. Higo, T. *et al.* Large magneto-optical Kerr effect and imaging of magnetic octupole domains in an antiferromagnetic metal. *Nat. Photonics* **12**, 73–78 (2018).
12. Radu, I. *et al.* Transient ferromagnetic-like state mediating ultrafast reversal of antiferromagnetically coupled spins. *Nature* **472**, 205–209 (2011).
13. Lovinger, D. J. *et al.* Magnetoelastic coupling to coherent acoustic phonon modes in the ferrimagnetic insulator GdTiO₃. *Phys. Rev. B* **102**, 1–10 (2020).
14. Kojima, E. *et al.* Observation of the spin-charge thermal isolation of ferromagnetic Ga_{0.94}Mn_{0.06}As by time-resolved magneto-optical measurements. *Phys. Rev. B - Condens. Matter Mater. Phys.* **68**, 1–4 (2003).

REVIEWER COMMENTS

Reviewer #1 (Remarks to the Author):

I have now read the comments of Reviewer #1 as well as the author's replies. I am very glad to see that my previous criticism motivated the authors to carry on a set of additional measurements which shed light on a coupling of phonon modes to the interlayer exchange coupling. Although I find that in the present version the manuscript is rather a specialized study, given its relevance to the studies interconnecting 2D magnetism and topological order I consider recommending publication in Nature Communication after the authors address my criticism provided below:

- My main criticism is related to the claims the authors provide in the manuscript. Using an external magnetic field and temperature authors provide a piece of strong evidence that the considered phonons are linked to the magnetism of MnBi₂Te₄. However, here and there the authors state the inverse process takes place, in which the phonon modulates the exchange coupling. Below I list several examples which I encountered reading the manuscript:

Page 1, line 28-29:

"A microscopic description of this phenomenon using density functional theory highlights the critical role played by phonons modulating the interlayer exchange coupling"

Page 2, line 18:

"In this work, we observe that interlayer magnetic order in MnBi₂Te₄ is strongly coupled to phonons..."

Is it experimentally verified that the phonons influence the magnetic order? I think the authors do not provide any shreds of evidences supporting this claim. The authors rather say:

"Nonetheless, it is clear from Fig. 4d that the coherent phonons track the AFM order parameter in accord with magnetophononic coupling."

and show that in both statics and dynamics phonons follow the magnetic order parameter.

What does the word "coherent" mean in defining magnetophononic coupling? Is it indeed coherent? The authors demonstrate no indications coupling the magnetic and phononic excitations, such as, for example, simultaneous excitation of coherent magnons and phonons or repulsion between the modes.

I thus kindly ask the authors to remove all the unjustified claims from the manuscript. I also suggest the authors consider changing the title to something saying "Evidence of magnetophononic coupling in MnBi₂Te₄": that is the way they refer to it in the abstract.

- Page 6, line 14-15:

"Instead, we propose that the momentum is provided by the AFM spin wave, via a magnetophononic wave-mixing process."

Are there any real indications of the AFM spin wave? Or simply the static AFM order plays a role?

- Page 11, line 7-9:

"Our experimental results then point to a mechanism of magnetophononic coupling at a sub-picosecond timescale that is similar to the generation of zone boundary phonons via magnetophononic wave-mixing."

What do the authors mean by saying that the magnetophononic coupling plays a role on the picosecond scale?

Can it be different given the frequency of the phonon (>1 THz) which is coupled to the magnetic order and seen in the MO Raman spectrum?

Do the authors want to say that the excitation of the phonon is mediated by the light-induced changes in the exchange interaction?

I thus would like the authors to clarify this statement.

- Given the relevance of the work to the 2D magnetism and light-induced control, I suggest the authors cite the latest time-resolved ultrafast studies on 2D antiferromagnetism in NiPS₃ the same family compound as FePS₃ the authors refer to: Nat. Comm., 12, 4837 (2021); Sci. Adv. 7, eabf3096 (2021).

Reviewer #3 (Remarks to the Author):

The paper by H. Padmanabhan et al. presents a study of MnBi₂Te₄ using a combination of magneto-Raman scattering, transient reflectivity, ultrafast electron diffraction, and density functional theory calculations. The authors claim that “the evolution of the spectral weight [in Raman] is consistent with the excitation of forbidden zone-boundary modes of the A_{1g}⁽¹⁾ and A_{1g}⁽²⁾ phonon branches.” The mechanism proposed by the authors is that magnetic cell doubling leads to folding of the phonon dispersion and activates modes that are originally forbidden in first-order Raman scattering (despite the absence of a structural phase transition). I believe that the observations are interesting, and the authors have appropriately addressed most of the comments from the previous round of review. However, I also think that the arguments supporting the proposed scenario are currently not very strong. For this reason, I cannot support acceptance of the paper in the present form. The paper may become suitable for Nature Communications after a thorough analysis to rule out other possible scenarios in the Raman data and clarification of several technical aspects that remain unaddressed.

Other possible scenarios in Raman are:

- 1) The authors fit the A_{1g}⁽¹⁾ mode to a Fano lineshape (which describes the interference between a continuum – electronic or phononic – and a discrete phonon resonance), while they fit A_{1g}⁽²⁾ to a Lorentzian lineshape. However, for A_{1g}⁽²⁾, the exact lineshape is very hard to establish because the mode overlaps to the E_g⁽³⁾ mode. It could be that mode A_{1g}⁽²⁾ also has a Fano lineshape, and the latter arises because of the interference with an electronic continuum. In MnBi₂Te₄ this continuum would stem from the residual carrier density around the Dirac surface states, which is gapped out upon time-reversal symmetry breaking (through the emergence of antiferromagnetic order or the application of a magnetic field). The authors should rule out that the residual intensity in Raman is not a trivial effect of the temperature and field dependence of this electronic continuum.
- 2) Another possibility is that the residual intensity is governed by a resonant Raman type of effect. The authors perform Raman using a laser at 632.8 nm, a wavelength that is resonant with interband transitions of MnBi₂Te₄: the temperature and magnetic field dependence of these interband transitions could provide resonant enhancement for the scattering from coupled phonons.
- 3) The residual intensity in Raman could come from second-order Raman scattering of bimagons that spectrally overlap with the detected phonons (especially A_{1g}⁽¹⁾). I find this explanation rather unlikely because of energetic constraints on mode A_{1g}⁽²⁾, but the authors should comment on this as well.

Moreover, the transient reflectivity experiments of Fig. 4b are performed under the same conditions of incident fluence on the sample, not of absorbed fluence. Can the authors rule out that the detected coherent phonon magnetic field dependence (Fig. 4e) does not stem from the field-induced change of the interband transition at 1.55 eV? As this process is nonlinear in nature (because it involves a complex lineshape change in an interband transition oscillator), the

normalization of the pump-probe traces may not be sufficient.

The authors complement their time-domain results with ultrafast electron diffraction measurements. However, the pump fluence used in these experiments is orders of magnitude larger than the one used for the transient reflectivity measurements. It is not clear to me how the authors can connect measurements performed under very different experimental conditions (most likely probing very different regimes) to discuss nonequilibrium phonon interactions.

Minor comments:

- In the introduction, the authors state "*A new paradigm has recently emerged with the discovery of magnetism in layered, quasi-two-dimensional materials.*" In this form, the statement is not correct, as the topic of magnetism in layered quasi-two-dimensional materials is instead very old (see the example of cuprate parent compounds, iridates, and other van der Waals coupled Mott insulators). The authors likely refer to atomically thin magnets, but this is also a topic disconnected to what the paper discusses (as the MnBi_2Te_4 material probed by the authors is in the bulk limit).

- I suggest that the authors include the labels of the E_g phonons in Figs. 1c-d, as readers can get confused about the features the authors are analyzing (the black arrow on top is not mentioned in the text and therefore not clear).

- The authors may want to avoid the term "spectral weight" when commenting the Raman scattering results. "Spectral weight" is a well-defined concept in optical conductivity measurements, and this may lead to confusion in some readers.

- At the end of the paper the authors may want to comment on sum-frequency ionic Raman scattering as an alternative way for realizing the coherent excitation of the Raman-active phonon modes.

- The authors should establish a comparison with J. Choe et al., Nano Lett. 21, 6139 (2021).

Response to Reviewer Comments

NCOMMS-21-13534A: Coherent magnetophononic coupling in MnBi_2Te_4

Contents

i.	Response summary	...	2
ii.	Point-by-point response – Reviewer #1	...	3
iii.	Point-by-point response – Reviewer #3	...	6

**Response summary**

We sincerely thank the reviewers for their careful reading of our manuscript and for providing constructive
criticism. With the goal of addressing the reviewers' comments on our work, we have carried out some
targeted experiments. These are –

(To rule out resonant Raman artifacts as an explanation for the observed phenomena)

- • Raman spectroscopy at three different laser excitation energies spanning 1.5 to 2.5 eV.
• Raman spectra as a function of temperature across the AFM transition, at a new laser excitation
energy (1.56 eV or 785 nm)

(To rule out magnons as an explanation for the observed phenomena)

- • Polarized Raman spectra as a function of temperature across the AFM transition.

Several changes have been made to the manuscript, including the results of the above experiments,
additional analysis as requested by the reviewers, and other smaller corrections addressing the reviewers'
concerns. Some of these are –

- • Two new sections (S5, S6) in the Supplementary Information.
• Three new figures (Figs. S5, S6, S7) in the Supplementary Information.
• Changes to the main text, highlighted in blue font.
• A modified title – 'Interlayer magnetophononic coupling in MnBi₂Te₄'.

We believe the revisions motivated by the reviewers' comments have strengthened the scientific content
and clarity of our manuscript. The following pages contain point-by-point responses to the reviewers.

Point-by-point response – Reviewer #1

I have now read the comments of Reviewer #1 as well as the author’s replies. I am very glad to see that
my previous criticism motivated the authors to carry on a set of additional measurements which shed light
on a coupling of phonon modes to the interlayer exchange coupling. Although I find that in the present
version the manuscript is rather a specialized study, given its relevance to the studies interconnecting 2D
magnetism and topological order I consider recommending publication in Nature Communication after
the authors address my criticism provided below:

We thank the reviewer for their careful reading of our manuscript as well as our response. We are happy
to hear that they find our work to be of relevance to the scientific community studying 2D magnetism and
topological phenomena. Below, we address the reviewers’ additional comments.

• My main criticism is related to the claims the authors provide in the manuscript. Using an external
magnetic field and temperature authors provide a piece of strong evidence that the considered phonons
are linked to the magnetism of MnBi₂Te₄. However, here and there the authors state the inverse process
takes place, in which the phonon modulates the exchange coupling. Below I list several examples which I
encountered reading the manuscript:

Page 1, line 28-29:

“A microscopic description of this phenomenon using density functional theory highlights the critical role
played by phonons modulating the interlayer exchange coupling”

Page 2, line 18:

“In this work, we observe that interlayer magnetic order in MnBi₂Te₄ is strongly coupled to phonons...”

Is it experimentally verified that the phonons influence the magnetic order? I think the authors do not
provide any shreds of evidences supporting this claim. The authors rather say: “Nonetheless, it is clear
from Fig. 4d that the coherent phonons track the AFM order parameter in accord with magnetophononic
coupling.” and show that in both statics and dynamics phonons follow the magnetic order parameter.

The reviewer is correct – we have not provided direct experimental evidence that phonons modulate
exchange interactions in MnBi₂Te₄. Our assertion that A_{1g} phonons modulate the interlayer exchange
coupling instead comes from our theoretical model are in full agreement with the experimental results. In
this sense, this is a derived result by combining theory and experiments, but nonetheless, we believe, a
robust conclusion.

We note that the phenomenology of this model is quite general – the crystal momentum required for the
excitation of zone-boundary phonons along the crystallographic c-axis is provided by the AFM order
along the same direction, via the free energy term written in Equation 2 in the main text. This necessarily
requires the modulation of the interlayer exchange coupling J^{\perp} by the phonons under consideration. The
density functional theory calculations further flesh out the microscopic details of this model, and as
shown in Fig. 3e, exhibit excellent agreement with our experimental results.

Regardless, since the above claim is based on theoretical modeling of our experimental results rather than
direct experimental evidence, as per the reviewer’s request, we have re-framed all instances of such
claims in the abstract and main text to appropriately reflect this, highlighted in blue. The changes are in –

- - page 1, line 29
- page 3, line 1

- page 8, line 8

What does the word “coherent” mean in defining magnetophononic coupling? Is it indeed coherent? The
authors demonstrate no indications coupling the magnetic and phononics excitations, such as, for
example, simultaneous excitation of coherent magnons and phonons or repulsion between the modes.

I thus kindly ask the authors to remove all the unjustified claims from the manuscript. I also suggest the
authors consider changing the title to something saying “Evidence of magnetophononic coupling in
MnBi₂Te₄”: that is the way they refer to it in the abstract.

Our use of the word coherent here was due to the spatial coherence of the antiferromagnetic order
required to observe the experimental phenomena described in Figs. 1 and 2. Our model of
magnetophononic wave-mixing relies on this coherence as well. However, we agree that the word
‘coherent’ may be misunderstood as the simultaneous excitation of coherent magnons and phonons, hence
we have removed it from the title. We have now changed the title to ‘Interlayer magnetophononic
coupling in MnBi₂Te₄’, to emphasize the role of interlayer AFM coupling.

• Page 6, line 14-15:

“Instead, we propose that the momentum is provided by the AFM spin wave, via a magnetophononic
wave-mixing process.”

Are there any real indications of the AFM spin wave? Or simply the static AFM order plays a role?

The phrase ‘AFM spin wave’ in our main text simply refers to the static AFM order. However, as pointed
out by the reviewer, we realize that the phrase ‘spin wave’ is usually understood to refer to either a
magnon or spin precession. We have now modified all instances of ‘AFM spin wave’ in the manuscript to
instead say ‘AFM order’.

• Page 11, line 7-9:

“Our experimental results then point to a mechanism of magnetophononic coupling at a sub-picosecond
timescale that is similar to the generation of zone boundary phonons via magnetophononic wave-mixing.”

What do the authors mean by saying that the magnetophononics coupling plays a role on the picosecond
scale?

Can it be different given the frequency of the phonon (>1 THz) which is coupled to the magnetic order
and seen in the MO Raman spectrum?

Do the authors want to say that the excitation of the phonon is mediated by the light-induced changes in
the exchange interaction?

I thus would like the authors to clarify this statement.

The sentence from our manuscript quoted by the reviewer is meant to convey the following –

- - The displacive phonon generation occurs at a sub-picosecond timescale.
- This displacive excitation can be described by a Raman-like process as outlined in page 11, line
15.
- The similarity of the results in Fig. 4e and Fig. 2 thus suggests that the same mechanism that
enables the excitation of zone-boundary phonons in our static Raman experiments also enables

the generation of zone-boundary phonons in our pump-probe experiments – namely
magnetophononic wavemixing. Since the displacive phonon generation occurs at a sub-
picosecond timescale, this implies a magnetophononic wave-mixing occurring at these timescales
as well.

We are not implying any light-induced change in the exchange coupling. We are instead simply referring
to the fact that the Raman-like displacive process allows for zone-boundary phonon excitation by the
same mechanism as in the static Raman – crystal momentum provided by the AFM order, mixing with the
phonon wavevector.

The reviewer is correct – since the observed phonons have frequencies >1 THz, the phonons excited in
the pump-probe measurements (including our proposed zone-boundary modes) must necessarily occur at
sub-picosecond timescales.

We realize the wording of the sentence quoted by the reviewer is ambiguous. Hence, we have re-framed it
(page 11, line 17) to communicate the above arguments with more clarity.

• Given the relevance of the work to the 2D magnetism and light-induced control, I suggest the authors
cite the latest time-resolved ultrafast studies on 2D antiferromagnetism in NiPS₃ the same family
compound as FePS₃ the authors refer to: Nat. Comm., 12, 4837 (2021); Sci. Adv. 7, eabf3096 (2021).

We thank the reviewer for pointing out these papers. These are now cited in the Discussion section of the
manuscript (page 14, line 23)

**Point-by-point response – Reviewer #3**

The paper by H. Padmanabhan et al. presents a study of MnBi₂Te₄ using a combination of magneto-
Raman scattering, transient reflectivity, ultrafast electron diffraction, and density functional theory
calculations. The authors claim that “*the evolution of the spectral weight [in Raman] is consistent with the*
*excitation of forbidden zone-boundary modes of the A_{1g}⁽¹⁾ and A_{1g}⁽²⁾ phonon branches.*” The mechanism
proposed by the authors is that magnetic cell doubling leads to folding of the phonon dispersion and
activates modes that are originally forbidden in first-order Raman scattering (despite the absence of a
structural phase transition). I believe that the observations are interesting, and the authors have
appropriately addressed most of the comments from the previous round of review.

We are happy to hear that the reviewer finds our experimental observations and proposed theoretical
model interesting and thank them for their careful and critical comments on our manuscript.

However, I also think that the arguments supporting the proposed scenario are currently not very strong.
For this reason, I cannot support acceptance of the paper in the present form. The paper may become
suitable for Nature Communications after a thorough analysis to rule out other possible scenarios in the
Raman data and clarification of several technical aspects that remain unaddressed.

We have carried out targeted experiments to address the reviewer’s comments and provide further
experimental evidence to support our claims. Below, we outline the results of our new experiments,
thorough analysis thereof, and clarify other technical questions raised by the reviewer.

Other possible scenarios in Raman are:

1) The authors fit the A_{1g}⁽¹⁾ mode to a Fano lineshape (which describes the interference between a
continuum – electronic or phononic – and a discrete phonon resonance), while they fit A_{1g}⁽²⁾ to a
Lorentzian lineshape. However, for A_{1g}⁽²⁾, the exact lineshape is very hard to establish because the mode
overlaps to the E_g⁽³⁾ mode. It could be that mode A_{1g}⁽²⁾ also has a Fano lineshape, and the latter arises
because of the interference with an electronic continuum. In MnBi₂Te₄ this continuum would stem from
the residual carrier density around the Dirac surface states, which is gapped out upon time-reversal
symmetry breaking (through the emergence of antiferromagnetic order or the application of a magnetic
field). The authors should rule out that the residual intensity in Raman is not a trivial effect of the
temperature and field dependence of this electronic continuum.

While the Dirac surface states are indeed expected to be gapped out upon the breaking of time-reversal
symmetry, we note that this is an effect that only occurs at the surface. On the other hand, the 1.96 eV
(633 nm) laser excitation used in our Raman experiments have a penetration depth of over 20 nm¹, thus
probing the bulk. Our measurements were done on bulk crystals. The Dirac surface states are present only
in the first few septuple layers² and would thus have a negligible overall contribution to our experiments.

Furthermore, the as-grown crystals of MnBi₂Te₄ used in our work³ (and other studies) exhibit significant
defect doping, such that the Fermi level is 0.2 eV above the conduction band minimum². Any effects in
our measurements due to an electronic continuum, including on the A_{1g}⁽¹⁾ mode, would likely be due to
bulk conduction band states. We did not observe any change in the continuum Raman scattering either as
a function of magnetic field or temperature.

2) Another possibility is that the residual intensity is governed by a resonant Raman type of effect. The
authors perform Raman using a laser at 632.8 nm, a wavelength that is resonant with interband transitions

of MnBi_2Te_4 : the temperature and magnetic field dependence of these interband transitions could provide
resonant enhancement for the scattering from coupled phonons.

Based on the above comments, we carried out Raman measurements as a function of laser excitation
energy to rule out resonant Raman effects as an explanation for the phenomena observed in Figs. 1 and 2.
Our results are shown in Fig. S5, for laser excitation energies of 1.58 eV, 1.96 eV, and 2.71 eV. It is
observed that phonon peak intensities indeed change as a function of the excitation energy, however these
changes occur across all the detected Raman phonon modes. In contrast, the temperature- and field-
dependent magnetophononic effects highlighted in Figs. 1 and 2 are observed only in the $A_{1g}^{(1)}$ and $A_{1g}^{(2)}$
modes, with negligible changes in the scattering intensities of other modes. Our observations are thus
inconsistent with resonant Raman effects.

Furthermore, we confirm that the scattering intensity of the $A_{1g}^{(2)}$ mode exhibits the same dependence
across the AFM transition with the 1.58 eV (785 nm) excitation (Fig. S6), as it does in the original results
reported in our manuscript, obtained with a 1.96 eV (633 nm) excitation. This is once again consistent
with an effect arising from the AFM order, as described in our model of magnetophononic wave-mixing.

Finally, we highlight the fact that the anomalous scattering intensity observed in the $A_{1g}^{(1)}$ and $A_{1g}^{(2)}$ modes
*quantitatively* track the AFM order, rather than just magnetic order in general. This is highlighted in Fig.
2b, where the anomalous scattering intensity disappears in the FM phase.

Based on the above arguments, and further energetic arguments detailed in Supplementary Information
Section S5, resonant Raman effects may be ruled out as the origin of the phenomena observed in Figs. 1
and 2. We have included additional text in page 5, line 15 of the main text pointing to this section in the
Supplementary Information.

3) The residual intensity in Raman could come from second-order Raman scattering of bimagnons that
spectrally overlap with the detected phonons (especially $A_{1g}^{(1)}$). I find this explanation rather unlikely
because of energetic constraints on mode $A_{1g}^{(2)}$, but the authors should comment on this as well.

We first note the magnon energies measured using inelastic neutron scattering experiments⁴ – zone-center
magnons are at around 1 meV (~8 meV) and zone-boundary magnons have a maximum energy of 3 meV
(~25 cm^{-1}). As pointed out by the reviewer, this rules out the possibility of one-magnon resonances
interfering with either of the phonon peaks, and also rules out the possibility of a two-magnon resonance
interfering with the $A_{1g}^{(2)}$ peak at 115 cm^{-1} .

Regardless, we carried out polarization-dependent Raman measurements across the AFM transition to
confirm this. Our results, discussed in Section S6 and shown in Fig. S7 show that the anomalous
scattering intensity coincident with the $A_{1g}^{(2)}$ zone-center phonon has an A_{1g} symmetry, ruling out a
magnon, which would be expected to have off-diagonal terms, and thus have an E_g symmetry.
Furthermore, we do not observe any signatures of a broad magnetic continuum that is typically expected
of two-magnon Raman scattering.

Based on the above arguments, and the detailed discussion in Section S6, magnons may be ruled out as
the origin of the phenomena observed in Figs. 1 and 2. We have included additional text in page 5, line 17
of the main text pointing to this section in the Supplementary Information.

Moreover, the transient reflectivity experiments of Fig. 4b are performed under the same conditions of
incident fluence on the sample, not of absorbed fluence. Can the authors rule out that the detected

coherent phonon magnetic field dependence (Fig. 4e) does not stem from the field-induced change of the
interband transition at 1.55 eV? As this process is nonlinear in nature (because it involves a complex
lineshape change in an interband transition oscillator), the normalization of the pump-probe traces may
not be sufficient.

We start by noting that changes in the pump-probe response as function of magnetic field may have
broadly two origins – (i) changes in optical absorption, and (ii), other qualitative changes in the bands
associated with the optical transition that can affect the el-ph coupling.

We understand that the reviewer is concerned with point (i) above. While it is true that magnetic field-
dependent measurements at a fixed fluence do not guarantee a fixed absorbed fluence, we would like to
highlight that the normalization procedure used in Fig. 4 specifically accounts for this. That is, the results
in Fig. 4 are effectively normalized to the *absorbed* fluence, using the initial pump-probe reflectivity
amplitude as a marker of the absorbed fluence.

The above approach is validated by showing in Section S7 that the initial pump-probe reflectivity
amplitude varies linearly with fluence at a *fixed* magnetic field. Our normalization procedure thus
accounts for point (i) above, i. e. the difference in absorption as a function of magnetic field.

We have revised the sentences in page 11, line 1 in the main text and page 10, line 4 in the Supplementary
Information to communicate this normalization procedure with more clarity.

Point (ii) above has to do with changes in the band structure due to the magnetic field. We note that the
magnetic field-driven AFM \rightarrow FM phase transition only changes the interlayer magnetic order, while
leaving the in-plane magnetic order unchanged. The interlayer exchange coupling is an order-of-
magnitude weaker than the in-plane exchange coupling of 0.12 meV. Based on these energetic
considerations, we expect qualitative electronic structure changes due to out-of-plane magnetic fields to
be small.

Any such changes would likely be described by a magnetodielectric effect of the form $\chi_e = \chi_e^{(0)} + \gamma M^2$,
where χ_e is the electrical susceptibility, γ is magnetodielectric coefficient, and M is the net
magnetization. We explore possible changes to electron-phonon interactions (and thus Raman
susceptibility) due to such an effect in Supplementary Information Section S7 (page 11, line 4). To lowest
order, we show that such an effect can give rise to changes in the Raman susceptibility proportional to $\frac{d\chi_m}{du}$
and $\frac{dy}{du}$, which are the phonon modulation (described by displacement u) of the magnetic susceptibility χ_m
and magnetodielectric coupling coefficient γ , respectively. Field-dependent changes in the coherent
phonon amplitude due to these terms would then be a form of *indirect* magnetophononic coupling. Based
on our current pump-probe experimental data, we cannot completely rule out that such indirect
magnetophononic effects also have a contribution, in addition to the direct magnetophononic effects
highlighted in our manuscript.

Finally, we note that the coherent phonon amplitudes in Fig. 4 track the AFM order, which is indeed
consistent with the static Raman results and our model of direct magnetophononic wave-mixing – i. e.
zone-boundary phonons modes excited due to the crystal momentum of the AFM order.

We have added a sentence in page 13, line 21 to mention the form of a possible magnetodielectric effect
giving rise to an indirect magnetophononic coupling in the main text pointing to Section S7 in the
Supplementary Information.

The authors complement their time-domain results with ultrafast electron diffraction measurements.
However, the pump fluence used in these experiments is orders of magnitude larger than the one used for
the transient reflectivity measurements. It is not clear to me how the authors can connect measurements
performed under very different experimental conditions (most likely probing very different regimes) to
discuss nonequilibrium phonon interactions.

The reviewer is correct that the UED measurements were carried out at a much higher pump fluence. In
general, a higher pump fluence results in the excitation of a larger population of phonons, and thus results
in faster phonon thermalization than at lower fluences. The high-fluence UED measurements thus set a
lower bound for the timescale for phonon thermalization, as described in page 12, line 18 of the
Supplementary Information. The phonon thermalization timescale at the low fluences used in our optical
pump-probe measurements would be even longer than the timescale established using UED, further
supporting our assertion that the phonon subsystem remains in a nonequilibrium state throughout the
range of measured time delays.

Minor comments:

- In the introduction, the authors state “*A new paradigm has recently emerged with the discovery of*
*magnetism in layered, quasi-two-dimensional materials.*” In this form, the statement is not correct, as the
topic of magnetism in layered quasi-two-dimensional materials is instead very old (see the example of
cuprate parent compounds, iridates, and other van der Waals coupled Mott insulators). The authors likely
refer to atomically thin magnets, but this is also a topic disconnected to what the paper discusses (as the
MnBi_2Te_4 material probed by the authors is in the bulk limit).

We have revised the sentence to accurately reflect the state of the current literature.

- I suggest that the authors include the labels of the E_g phonons in Figs. 1c-d, as readers can get confused
about the features the authors are analyzing (the black arrow on top is not mentioned in the text and
therefore not clear).

Fig. 1c now includes an arrow pointing to the $E_g^{(3)}$ mode.

- The authors may want to avoid the term “spectral weight” when commenting the Raman scattering
results. “Spectral weight” is a well-defined concept in optical conductivity measurements, and this may
lead to confusion in some readers.

We have changed all instances of ‘spectral weight’ to instead read ‘scattering intensity’.

- At the end of the paper the authors may want to comment on sum-frequency ionic Raman scattering as
an alternative way for realizing the coherent excitation of the Raman-active phonon modes.

We have included a brief comment on sum-frequency ionic Raman scattering as a way to realize
excitation of Raman phonon modes, including a reference on such work in Bi_2Se_3 in page 14, lines 20 and
23.

- The authors should establish a comparison with J. Choe et al., Nano Lett. 21, 6139 (2021).

We have added text in page 4, lines 2 and 8 in the main text and page 6, line 11 of the Supplementary
Information to establish a comparison with the above paper.

REVIEWERS' COMMENTS

Reviewer #1 (Remarks to the Author):

I am happy to see that the authors satisfactorily addressed all my previous criticism. Therefore I am now recommending the paper to be published in Nature Communications.

Reviewer #3 (Remarks to the Author):

I found the revised manuscript substantially improved compared to the previous version. The authors provided satisfactory answers to all my comments and added new data to rule out alternative scenarios. I only suggest that the authors modify the presentation of the UED data, justifying the inclusion of a high fluence measurement while the rest of the ultrafast experiments are performed at low fluence. This would help to avoid confusion in the readers. I believe that the paper will then be fit for publication in Nature Communications.